# PO-DREAMER: MEMORY GUIDED WORLD MODELS FOR PARTIALLY OBSERVABLE REINFORCEMENT LEARNING

## ABSTRACT

World models predict future states and rewards by learning compact state representations of the environment, thereby enabling efficient policy optimization. World-model-based reinforcement learning (RL) algorithms have demonstrated significant advantages in complex tasks. However the scenarios in real world application are always partially observable (i.e., image based RL and multi-agent RL), and contain non-stationary dynamics. To address the challenges in Partially Observable Markov Decision Processes (POMDP) scenarios, we propose a novel memory guided world model named PO-Dreamer. Besides current observation, we adaptively extract meaningful cues from memory which is helpful to model the environmental dynamics. Then, the features of current observation and memory are fused by the fusion mechanism to predict state transition and future rewards. Extensive experiments on both single-agent (Atari 100K) and multi-agent (SMAC) tasks demonstrate that our method achieves state-of-the-art (SOTA) performance compared to existing strong baselines.

## 1 INTRODUCTION

Model-based reinforcement learning (MBRL) Moerland et al. (2023); Luo et al. (2022) enhances sample efficiency through the explicit construction of environment dynamics models. Known as world models Ha & Schmidhuber (2018), these environment models learn compact state representations and predict rewards, effectively capturing the environment's dynamics. World models enable agents to improve their behavior by operating within the imagined latent space, significantly reducing the dependence on expensive real-world interactions.

Although world models demonstrate remarkable sample efficiency in complex control tasks with high-dimensional observations Schrittwieser et al. (2020); Hafner et al. (2025), real-world environments are often partially observable. Such environments are often modeled as Partially Observable Markov Decision Processes (POMDPs) Kaelbling et al. (1998), which exhibit non-stationary characteristics Zhu et al. (2023), especially in complex dynamic systems or multi-agent scenarios. That is, the same observations may receive different observation transitions and rewards under the same action. The reason is that an observation may correspond to different states in a POMDP. This characteristic poses a bottleneck in applying MBRL to the real world. Two illustrative examples are presented in Fig. 1. In the Atari Bellemare et al. (2013) *Breakout* scenario, the agent relying solely on the current observations can detect the instantaneous positions of the paddle and ball, while critical dynamics like the ball's velocity vector remain unobserved. Similarly, in the StarCraft Multi-Agent Challenge (SMAC) Samvelyan et al. (2019) *2m_vs_1z* scenario, decentralized agents receive only static unit positions from single-frame observations, lacking critical motion vectors and action intentions. Such partial observability poses significant challenges for environment modeling and RL. To address this limitation, a promising solution involves leveraging memory to construct a robust world model that effectively integrates past experiences and dynamically adapts to environmental changes. For instance, agent can infer the velocity and direction of the ball from consecutive frames in Atari Breakout, while SMAC agents can predict other agents' strategies by leveraging relevant or similar past interactions retrieved from memory. This approach is both highly effective and intuitive, as it mirrors how humans extract critical information from real-world environments by drawing relevant past experiences from memory.

Due to the predictive nature, world models inherently rely on sequential networks. Most existing approaches Hafner et al. (2019a; 2020; 2023; 2025); Micheli et al. (2022); Chen et al. (2022); Zhang et al. (2023); Robine et al. (2023) adopt Recurrent Neural Networks (RNN) Zaremba et al. (2014) or Transformer Vaswani et al. (2017) as their architectural backbone, and historical information has been involved. However, these methods suffer from two limitations: Firstly, existing approaches typically only process memory information through native and direct mechanisms. Consequently, these valuable memory traces remain underutilized, and the challenge of how to adaptively extract the most valuable cues from them has not been fully explored. Secondly, existing methods only focus on the recent history in the same episode, and the valuable but far memory is ignored. Hence, we propose a novel algorithm called PO-Dreamer, a memory guided world model for POMDPs. Specifically, our architecture introduces an additional memory encoder as well as the convention encoder Hafner et al. (2023; 2025), which focuses on memory and current observation respectively. For the memory encoder, the historical observations and actions of the current and past episodes are constructed as memory firstly. Then, the current observation provides the query, while the observations and actions from memory compute the key and value. The multi-head attention Vaswani et al. (2017) is used to extract key cues from the memory. Since the unobservable parts are always non-deterministic and the collected samples are not complete, hence we model the memory features as a learnable Gaussian distribution by a variational autoencoder (VAE) Kingma et al. (2013); Rezende et al. (2014), which could avoid fast overfitting and enhance the robustness of PO-Dreamer against the environment's non-stationarity. Based on two encoders, the fused sequence model and dynamics predictor are performed to jointly construct an implicit representation of state, and further used to better predict the state transitions and future rewards.

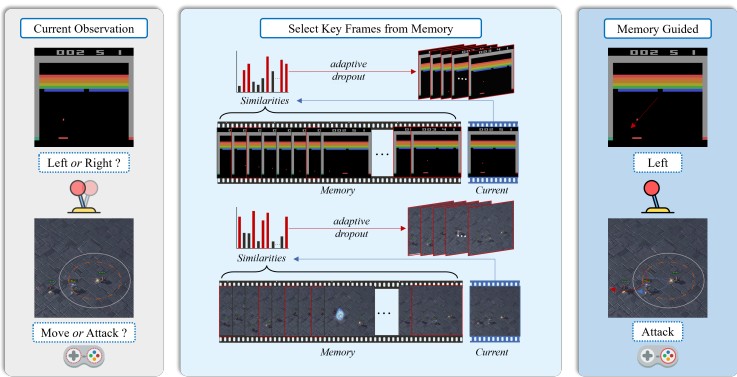

Figure 1: Visualization of the Atari *Breakout* and the SMAC *2m_vs_ 1z*, decision-making based on the current observation is challenging due to the environment is partially observable. However, extracting key information from memory can effectively assist the agent in making decisions.

The key contributions of our work are as follows:

- A memory guided world model, PO-Dreamer, is proposed to extract key temporal information to infer unobservable aspects of the environment, effectively addressing the challenges of non-stationarity in POMDPs.
- A novel fusion mechanism is designed to integrate memory and current observations, enabling a more comprehensive modeling of environmental dynamics.
- PO-Dreamer could be used for both single-agent and multi-agent tasks. Experiments on Atari and SMAC both demonstrate that our method offers significant advantages in both performance and sample efficiency.

## 2 RELATED WORK

### 2.1 MODEL BASED REINFORCEMENT LEARNING (MBRL)

MBRL improves the sample efficiency and planning capabilities of RL by explicitly learning a model of the environment dynamics Sutton (1991); Chua et al. (2018); Janner et al. (2019), which

are also referred to world models. The development of PlaNet Hafner et al. (2019b) made a significant advancement through the Recurrent State Space Model, enabling policy learning within compact latent space representations. This work was extended by the Dreamer series Hafner et al. (2019a; 2020; 2023; 2025), which systematically developed the paradigm of "world model learning + trajectory imagination + policy optimization." MuZero Schrittwieser et al. (2020) advanced the field by learning implicit environment and value representations combined with Monte Carlo Tree Search for planning. Based on Dreamer, recent methods have incorporated various architectures for world modeling to improve effectiveness for complex environments, such as Transformer Micheli et al. (2022); Zhang et al. (2023); Robine et al. (2023) or Diffusion Alonso et al. (2024). Compared with these methods, we primarily focus on POMDPs and aim to infer the unobservable information from memory. A novel memory encoder is introduced to extract key clues adaptively using the attention mechanism. Consequently, our approach demonstrates superior performance in POMDPs settings, including image-based single agent RL and multi-agent RL. Although ISO-Dream Pan et al. (2022) similarly employs a multi-branch world model, our approaches are fundamentally different in motivation. ISO-Dream operates on the assumption that environments contain decision-irrelevant information, and thus aims to decouple controllable from non-controllable dynamics in observations. In contrast, our method tackles POMDPs by explicitly reconstructing latent states through memory. Furthermore, several studies have extended world models to multi-agent learning scenarios. For example, MAMBA Egorov & Shpilman (2022) employs attention mechanisms to enable information sharing among multi-agents' world models. Building upon this foundation, MAG Wu et al. (2023) enhances predictive capability by integrating Model Predictive Control (MPC) for multi-step state forecasting, while MACD Venugopal et al. (2023) utilizes latent variable world model to separately model global states and local observations. Unlike these methods requiring agent communication during world model training, our approach maintains decentralized world model for each agent. This improves our applicability in realistic scenarios.

## 2.2 History used for POMDPs

The effectiveness of historical observation utilization in POMDPs has been established in prior works. DRQN Hausknecht & Stone (2015) enhanced DQN Mnih et al. (2015) by incorporating LSTM networks to process sequential observations. Building upon this foundation, DDRQN Foerster et al. (2016) extends this approach to multi-agent scenarios and ADRQN Zhu et al. (2017) further integrates action information into the agent's historical observation sequence. LSTM-TD3 Meng et al. (2021) augments TD3 Fujimoto et al. (2018) for continuous control tasks. DTQN Esslinger et al. (2022) replaces traditional LSTM with Transformer to encode the agent's historical observation information. In multi-agent setting, the sequential models such as RNNs are often chosen as the backbone Foerster et al. (2018); Rashid et al. (2020). Especially in a decentralized multi-agent learning setting, the history trajectory is also widely used to overcome the non-stationarity Zhai et al. (2023); Hu et al. (2023); Pritz & Leung (2025). R2I Samsami et al. (2024) integrates State Space Models (SSMs) into the world model to capture dependencies in long sequences. WorldMem Xiao et al. (2025) addresses the challenges of long-term 3D consistency in world simulation by introducing a memory bank with a state-based attention mechanism. Our PO-Dreamer differs from the above methods in two key aspects: Firstly, the memory information utilized in our method exhibits cross-episode persistence, encompassing not only the historical trajectory of the current episode but also integrating long-term memory from past interactions. Secondly, these methods merely process the current observation or historical information in isolation, without effectively designing a mechanism to leverage valuable information from memory based on the current observation.

## 3 Method

Considering the image-based RL task and multi-agent RL task as POMDPs, our method contains two integrated components: world modeling and behavior learning. The world modeling component employs a fusion architecture to predict state transitions and rewards from partial observations. Subsequently, the behavior learning is adopted on this learned model, which is applicable to both single-agent and multi-agent RL scenarios. MBRL learns a policy to maximize the final returns within the world model, which predicts the dynamics and reward function of the environment.

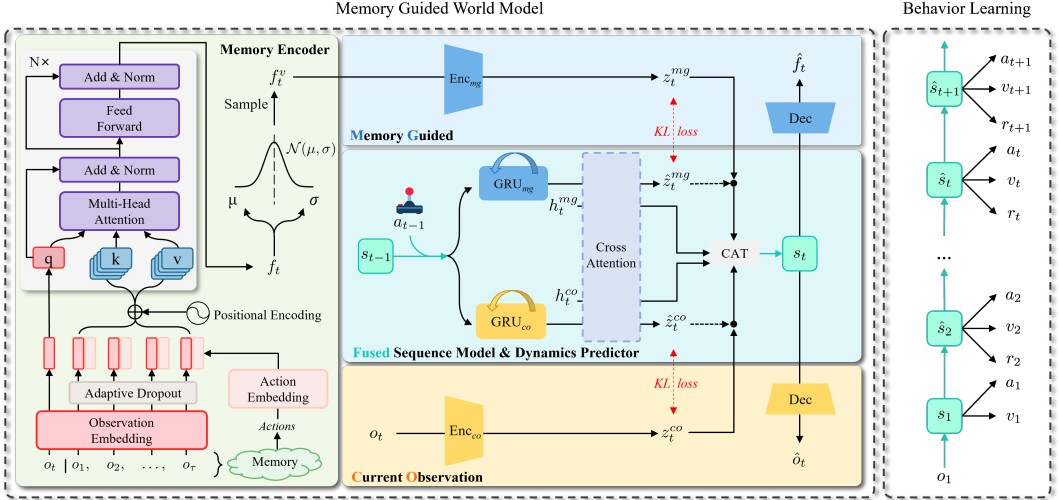

Figure 2: The architecture of our PO-Dreamer. The framework consists of two key components: the world model learning and behavior learning. The former employs dual branches to extract critical features from both current observations and memory, learning environment dynamics and reward functions. The latter performs RL based on the world model.

## 3.1 World Model Learning for POMDPs

POMDP is formally defined by a 7-tuple $\langle S, \mathcal{A}, \mathcal{O}, T, Z, R, \gamma \rangle$. $S$, $\mathcal{O}$, and $A$ denote the state space, observation space, and action space respectively. The dynamics function $T : S \times \mathcal{A} \to S$ specifies the state transition dynamics as $p(s_{t+1} \mid s_t, a_t)$, the observation function $Z : S \to \mathcal{O}$ describes the observation probability $p(o_t \mid s_t)$, the reward function $R : S \times \mathcal{A} \to R$ computes the immediate rewards, and $\gamma \in [0, 1]$ is the discount factor. Based on RSSM architecture Ha & Schmidhuber (2018), our world model takes the memory and current observation into account and aims to achieve the collaboration between them. Suppose the encoding of current observation and memory at time $t$ are $z_t^{co}$ and $z_t^{mg}$ respectively, and they could generate the recurrent state $h_t^{co}$ and $h_t^{mg}$. We argue that $z_t^{co}$ and $h_t^{co}$ correspond to the current observation, while $z_t^{mg}$ and $h_t^{mg}$ represent the memory of the state. And the latent state of the world model can be represented as $s_t \doteq \text{CAT}(h_t^{co}, z_t^{co}, h_t^{mg}, z_t^{mg})$. Considering $z_t^{co}$ and $z_t^{mg}$ are calculated from $h_t^{co}$ and $h_t^{mg}$ by the dynamics predictor, the state transition could be:

$$p(s_t|s_{t-1}, a_{t-1}) = p(h_t^{co}, z_t^{co}, h_t^{mg}, z_t^{mg}|h_{t-1}^{co}, z_{t-1}^{co}, h_{t-1}^{mg}, z_{t-1}^{mg}, a_{t-1})$$
$$= p(z_t^{co}, z_t^{mg}|h_t^{co}, h_t^{mg})p(h_t^{co}, h_t^{mg}|h_{t-1}^{co}, z_{t-1}^{co}, h_{t-1}^{mg}, z_{t-1}^{mg}, a_{t-1}). \quad (1)$$

During the learning, the world model receives the observation $o_t$ at each time step $t$, with the memory $m_t = \{\hat{o}_{\hat{t}}, \hat{a}_{\hat{t}}\}_{\hat{t}=1}^{\tau}$ of previous observation and action sequences. Firstly, the observation encoder and the memory encoder are used to process the current observation and the memory respectively:

$$\text{Observation Encoder:} z_t^{co} \sim q_\phi(z_t^{co} \mid h_t^{co}, o_t), \quad \text{Memory Encoder:} z_t^{mg} \sim q_\phi(z_t^{mg} \mid h_t^{mg}, m_t). \quad (2)$$

We designed a fusion mechanism that integrates the features extracted by the two encoders: At each timestep $t$, the fused sequence model generates the recurrent state $h_t$ based on the previous recurrent state $h_{t-1}^{co}, h_{t-1}^{mg}$, stochastic representation $z_{t-1}^{co}, z_{t-1}^{mg}$ and action $a_{t-1}$. And the dynamics predictor infers the stochastic representation $\hat{z}_t^{co}, \hat{z}_t^{mg}$.

$$\text{Fused Sequence model:} \quad h_t^{co}, h_t^{mg} = f_\phi(h_{t-1}^{co}, z_{t-1}^{co}, h_{t-1}^{mg}, z_{t-1}^{mg}, a_{t-1}),$$
$$\text{Fused Dynamics predictor:} \quad \hat{z}_t^{co}, \hat{z}_t^{mg} \sim p_\phi(\hat{z}_t^{co}, \hat{z}_t^{mg} \mid h_t^{co}, h_t^{mg}). \quad (3)$$

The implementation of the observation encoder is the same as the encoder in Hafner et al. (2025), here we mainly describe the memory encoder, fused sequence model, and fused dynamics predictor.

### 3.1.1 Memory Encoder

Firstly, the current observation paired with a memory window of $\tau$ time steps is encoded through $f_o(\cdot)$ separately to generate $f_o(o_t)$ and a sequence of memory observation embedding $\{f_o(\hat{o}_{\hat{t}})\}_{\hat{t}=1}^{\tau}$, the $\hat{t}$ is the time step in the memory. The embedding of the current observation is set as $f_t^c = f_o(o_t)$.

During environmental interaction, an agent can accumulate memories spanning tens of thousands of steps. Using all this memory for training would be computationally intensive and time-consuming. Considering that most of the memory has weak relevance to the current state, we introduce an adaptive dropout mechanism to identify key frames from massive memory and perform attention computation based on these key frames. This mechanism dynamically selects the most relevant memory fragments by computing the cosine similarity $S$ between memory embeddings and the current observation embedding.

$$\mathcal{M}_{o_t} = \arg \max_{\{t_i\}_{i=1}^{\eta} \subseteq [1,\tau]} \sum_{i=1}^{\eta} S(f_o(o_t), f_o(\hat{o}_{\hat{t}_i})) = \{f_o(\hat{o}_{\hat{t}})\}_{\hat{t}=1}^{\eta} \qquad (4)$$

where $\mathcal{M}_{o_t}$ represents the subset of top-$\eta$ memory selected for the current observation $o_t$. It ensures that only the most relevant memory information is utilized for world modeling. Subsequently, an attention mechanism is employed to capture the dependencies between the current observation and the memory, thereby aiding the world model in constructing a more robust state transition function.

Simultaneously, the action sequence is encoded via $f_a(\cdot)$ separately, producing the action embeddings $\{f_a(\hat{a}_{\hat{t}})\}_{\hat{t}=1}^{\eta}$. In addition, the positional encoding of $\hat{t} \in \{1, \eta\}$ is computed as Vaswani et al. (2017), denoted as $\{Pos_{\hat{t}}\}_{\hat{t}=1}^{\eta}$. We construct the memory embeddings through element-wise addition of observation embeddings, action embeddings, and positional encoding as $f_{\hat{t}}^c = f_o(o_{\hat{t}}) + f_a(a_{\hat{t}}) + Pos_{\hat{t}}$ where $\hat{t} \in (1, \eta)$.

Suppose $f_t^c \in \mathbb{R}^{1 \times d}$, and $F^c \in \mathbb{R}^{\eta \times d}$ who concertante $\{f_{\hat{t}}^c\}_{\hat{t}=1}^{\eta}$ together. Since we focus on the current state, hence the current context embedding $f_t^c$ provides the query $Q$, while memory embeddings $F^c$ are used to compute the keys $K$ and values $V$. The Multi-head Attention (MHA) Vaswani et al. (2017) is used:

$$\begin{aligned} \text{MultiHead}(Q, K, V) &= \text{CAT}(\text{head}_1, \ldots, \text{head}_h)W^O, \\ \text{head}_i &= \text{Attention}(f_t^c W_i^Q, F_c W_i^K, F_c W_i^V), \\ \text{Attention}(Q, K, V) &= \text{softmax}\left(\frac{QK^T}{\sqrt{d_k}}\right)V. \end{aligned} \qquad (5)$$

where $W_i^Q$, $W_i^K$ and $W_i^V$ are the the weight matrices of queries, keys and values in the $ith$ head. Subsequently, $N$ blocks, where each contains a feed forward layer, a layer normalization and a residual connection, are employed to calculate $f_t$ from $\text{MultiHead}(Q, K, V)$.

Since the non-stationary nature of the POMDPs, we employ a Variational Autoencoder (VAE) Kingma et al. (2013); Rezende et al. (2014) to model the latent dynamics as a Gaussian distribution. It could make the state transition and reward prediction of our world model more robust. Specifically, the mean $\mu_\phi(f_t)$ and standard deviation $\sigma_\phi(f_t)$ are calculated based on $f_t$ using $\text{VAE}_\phi$, and then $f_t^v$ is sampled from $\mathcal{N}(\mu_\phi(f_t), \sigma_\phi(f_t))$. To make the gradient backpropagation available, the reparameterization trick Kingma et al. (2013) is used instead:

$$f_t^v = \mu_\phi(f_t) + \sigma_\phi(f_t) \cdot \epsilon, \quad where \quad \epsilon \sim \mathcal{N}(0, 1). \qquad (6)$$

Subsequently, the encoder $\text{Enc}_{mg}$ encodes $f_t^v$ into the latent representation $z_t^{mg}$, which is the final output of the memory encoder in Eq. 4.

### 3.1.2 FUSED SEQUENCE MODEL

The fused sequence model learns to predict the recurrent state sequence in the world model, which enables it to plan in imagination. The sequence model predicts $h_t^{co}$ and $h_t^{mg}$ from the last latent state $s_{t-1} \doteq \text{CAT}(h_{t-1}^{co}, z_{t-1}^{co}, h_{t-1}^{mg}, z_{t-1}^{mg})$ and the last action $a_{t-1}$. We implement the sequence model as GRUs Cho et al. (2014) with recurrent state $h_t^{co}$ and $h_t^{mg}$ respectively:

$$h_t^{co} = \text{GRU}_{co}(s_{t-1}, a_{t-1}), \ h_t^{mg} = \text{GRU}_{mg}(s_{t-1}, a_{t-1}). \qquad (7)$$

### 3.1.3 FUSED DYNAMICS PREDICTOR

Given $h_t^{co}$ and $h_t^{mg}$ produced by the fused sequence model, the fused dynamics predictor aims to calculate $\hat{z}_t^{co}$ and $\hat{z}_t^{mg}$ according to Eq. 4. To achieve this, a cross attention module (CA) Vaswani

et al. (2017) is introduced to make $h_t^{co}$ and $h_t^{mg}$ interact with each other:

$$h_t^{co'}, h_t^{mg'} = \text{CA}(h_t^{co}, h_t^{mg}). \tag{8}$$

After processing the fused recurrent states through the attention mechanism, we pass $h_t^{co'}$ and $h_t^{mg'}$ into the dynamics predictors to obtain the stochastic latent representations $\hat{z}_t^{co}$ and $\hat{z}_t^{mg}$:

$$\hat{z}_t^{co} \sim p_\phi(\hat{z}_t^{co} \mid h_t^{co'}), \ \hat{z}_t^{mg} \sim p_\phi(\hat{z}_t^{mg} \mid h_t^{mg'}). \tag{9}$$

### 3.1.4 LEARNING

The world model compresses the information into latent representation, we can reconstruct the observation to ensure informative representations for learning. Set the latent state as $s_t \doteq \text{CAT}(h_t^{co}, z_t^{co}, h_t^{mg}, z_t^{mg})$, the decoder and predictors are used. Besides, a memory decoder is designed to reconstruct memory information:

$$\begin{aligned} \text{Observation decoder:} \quad & \hat{o}_t \sim p_\phi(\hat{o}_t \mid s_t), \quad \text{Memory decoder:} \quad \hat{f}_t \sim p_\phi(\hat{f}_t \mid s_t), \\ \text{Reward predictor:} \quad & \hat{r}_t \sim p_\phi(\hat{r}_t \mid s_t), \quad \text{Continue predictor:} \quad \hat{c}_t \sim p_\phi(\hat{c}_t \mid s_t). \end{aligned} \tag{10}$$

We denote the categorical cross entropy loss as $\text{catxent}(\cdot)$, and the binary cross entropy loss as $\text{binxent}(\cdot)$, which follows the set of DreamerV3 Hafner et al. (2023; 2025). And then the predictive loss $\mathcal{L}_{\text{pred}}(\phi)$ is the sum of the prediction losses to contrast the prediction with ground truth:

$$\begin{aligned} \text{Observation loss:} \quad & \mathcal{L}_o = \|\hat{o}_t - o_t\|_2^2, \quad \text{Memory loss:} \quad \mathcal{L}_f = \text{catxent}(\hat{f}_t, f_t), \\ \text{Reward loss:} \quad & \mathcal{L}_r = \text{catxent}(\hat{r}_t, r_t), \quad \text{Continue loss:} \quad \mathcal{L}_c = \text{binxent}(\hat{c}_t, c_t). \end{aligned} \tag{11}$$

For the fused dynamics predictor, we aim to make the $\hat{z}_t$ and $z_t$ as close as possible, the KL loss is performed. Similar to DreamerV3 Hafner et al. (2023; 2025), the dynamics loss $\mathcal{L}_{dyn}$ and representation loss $\mathcal{L}_{rep}$ are employed with the stop-gradient operator $\text{sg}(\cdot)$:

$$\begin{aligned} \mathcal{L}_{\text{dyn}}(\phi) &\doteq \max(1, \text{KL}\big[\text{sg}(z_t^{co}) \parallel \hat{z}_t^{co}\big] + \text{KL}\big[\text{sg}(z_t^{mg}) \parallel \hat{z}_t^{mg}\big]), \\ \mathcal{L}_{\text{rep}}(\phi) &\doteq \max(1, \text{KL}\big[z_t^{co} \parallel \text{sg}(\hat{z}_t^{co})\big] + \text{KL}\big[z_t^{mg} \parallel \text{sg}(\hat{z}_t^{mg})\big]). \end{aligned} \tag{12}$$

Finally, the world model parameters $\phi$ are optimized end-to-end to minimize the total loss with hyperparameter $\beta_{\text{dyn}} = 0.5, \beta_{\text{rep}} = 0.1$, following the set of DreamerV3 Hafner et al. (2023; 2025):

$$\mathcal{L}(\phi) \doteq \mathbb{E}_{q_\phi}\left[\sum_{t=1}^{T}(\mathcal{L}_{\text{pred}}(\phi) + \beta_{\text{dyn}}\mathcal{L}_{\text{dyn}}(\phi) + \beta_{\text{rep}}\mathcal{L}_{\text{rep}}(\phi))\right]. \tag{13}$$

### 3.2 BEHAVIOR LEARNING

The behavior learning is based on the latent space $\hat{s}_t = \text{CAT}(h_t^{co}, \hat{z}_t^{co}, h_t^{mg}, \hat{z}_t^{mg})$ imagined by the world model. We conducted experimental validation in both multi-agent and single-agent scenarios.

In single-agent scenario, we adopt the Actor-Critic architecture of Hafner et al. (2023; 2025). The actor seeks to learn the policy which could maximize the cumulative return $R_t \doteq \sum_{\tau=0}^{\infty} \gamma^\tau r_{t+\tau}$, and the critic approximates the return distribution under the actor's policy.

$$\begin{aligned} \text{Actor:} \quad & a_t \sim \pi_\theta(a_t \mid \hat{s}_t) \\ \text{Critic:} \quad & R_t \sim v_\theta(R_t \mid \hat{s}_t) \doteq \mathbb{E}_\pi\left(\sum_{\tau=0}^{\infty} \gamma^\tau r_{t+\tau} \mid \hat{s}_t\right) \end{aligned} \tag{14}$$

In multi-agent scenario, each agent maintains an individual world model and shares the same network weights, and learns from the joint latent state sequences $\{\hat{s}_t^i\}_{i=1}^{n}$ across all $n$ agents. As for RL policy learning, we adapt the Multi-Agent Transformer (MAT) framework Wen et al. (2022) with two key components: (1) an attention-based mechanism enabling inter-agent communication through latent state, (2) autoregressive action generation that preserves decentralized execution.

Table 1: Results on the Atari 100k benchmark, along with human-normalized aggregate metrics. Bold values denote the best-performing methods. Our approach significantly outperforms other world model baselines in mean score over 5 random seeds.

| Game | Random | Human | SimPLE | TWM | IRIS | DreamerV3 | MuDreamer | STORM* | HarDream | DIAMOND* | Ours |
|---|---|---|---|---|---|---|---|---|---|---|---|
| Alien | 227.8 | 7127.7 | 616.9 | 674.6 | 420.0 | 959.0 | 951.0 | 983.6 | 890.0 | 744.1 | **1265.1** |
| Amidar | 5.8 | 1719.5 | 74.3 | 121.8 | 143.0 | 139.0 | 153.0 | 204.8 | 141.0 | **225.8** | 151.7 |
| Assault | 222.4 | 742.0 | 527.2 | 682.6 | 1524.4 | 706.0 | 891.0 | 801.0 | 1003.0 | **1526.4** | 678.1 |
| Asterix | 210.0 | 8503.3 | 1128.3 | 1116.6 | 853.6 | 932.0 | 1411.0 | 1028.0 | 1140.0 | **3698.5** | 1244.3 |
| BankHeist | 14.2 | 753.1 | 34.2 | 466.7 | 53.1 | 649.0 | 156.0 | 641.2 | **1069.0** | 19.7 | 958.2 |
| BattleZone | 2360.0 | 37187.5 | 4031.2 | 5068.0 | 13074.0 | 12250.0 | 12080.0 | 13540.0 | 16456.0 | 4702.0 | **18693.0** |
| Boxing | 0.1 | 12.1 | 7.8 | 77.5 | 70.1 | 78.0 | **96.0** | 79.7 | 80.0 | 86.9 | 89.8 |
| Breakout | 1.7 | 30.5 | 16.4 | 20.0 | 83.7 | 31.0 | 34.0 | 15.9 | 53.0 | **132.5** | 85.3 |
| ChopperCommand | 811.0 | 7387.8 | 979.4 | 1697.4 | 1565.0 | 420.0 | 808.0 | 1888.0 | 1510.0 | 1369.8 | **2011.6** |
| CrazyClimber | 10780.5 | 35829.4 | 62583.6 | 71820.4 | 59324.2 | 97190.0 | 96128.0 | 66776.0 | 82739.0 | **99167.8** | 96214.5 |
| DemonAttack | 152.1 | 1971.0 | 208.1 | 350.2 | **2034.4** | 303.0 | 553.0 | 164.6 | 203.0 | 288.1 | 572.3 |
| Freeway | 0.0 | 29.6 | 16.7 | 24.3 | 31.1 | 0.0 | 5.0 | **33.5** | 0.0 | 33.3 | 19.0 |
| Frostbite | 65.2 | 4334.7 | 236.9 | 1475.6 | 259.1 | 909.0 | 1652.0 | 1316.0 | 679.0 | 274.1 | **2033.7** |
| Gopher | 257.6 | 2412.5 | 596.8 | 1674.8 | 2236.1 | 3730.0 | 1500.0 | 8239.6 | **13043.0** | 5897.9 | 6049.9 |
| Hero | 1027.0 | 30826.4 | 2656.6 | 7254.0 | 7037.4 | 11161.0 | 8272.0 | 11044.3 | **13378.0** | 5621.8 | 12158.4 |
| Jamesbond | 29.0 | 302.8 | 100.5 | 362.4 | 462.7 | 445.0 | 409.0 | 509.0 | 317.0 | 427.4 | **525.7** |
| Kangaroo | 52.0 | 3035.0 | 51.2 | 1240.0 | 838.2 | 4098.0 | 4380.0 | 4208.0 | 5118.0 | 5382.2 | **5573.9** |
| Krull | 1598.0 | 2665.5 | 2204.8 | 6349.2 | 6616.4 | 7782.0 | **9644.0** | 8412.6 | 7754.0 | 8610.1 | 8261.6 |
| KungFuMaster | 258.5 | 22736.3 | 14862.5 | 24554.6 | 21759.8 | 21420.0 | **26832.0** | 26182.0 | 22274.0 | 18713.6 | 25173.8 |
| MsPacman | 307.3 | 6951.6 | 1480.0 | 1588.4 | 999.1 | 1327.0 | 2311.0 | **2673.5** | 1681.0 | 1958.2 | 2481.3 |
| Pong | -20.7 | 14.6 | 12.8 | 18.8 | 14.6 | 18.0 | 18.0 | 11.3 | 19.0 | 20.4 | **20.5** |
| PrivateEye | 24.9 | 69571.3 | 35.0 | 86.6 | 100.0 | 882.0 | 1042.0 | **7781.0** | 2932.0 | 114.3 | 1109.4 |
| Qbert | 163.9 | 13455.0 | 1288.8 | 3330.8 | 745.7 | 3405.0 | 4061.0 | **4522.5** | 3933.0 | 4499.3 | 4325.9 |
| RoadRunner | 11.5 | 7845.0 | 5640.6 | 9109.0 | 9614.6 | 15565.0 | 8460.0 | 17564.0 | 14646.0 | 20673.2 | **22518.2** |
| Seaquest | 68.4 | 42054.7 | 683.3 | 774.4 | 661.3 | 618.0 | 428.0 | 525.2 | 665.0 | 551.2 | **815.5** |
| UpNDown | 533.4 | 11693.2 | 3350.3 | 15981.7 | 3546.2 | 9234.0 | 26494.0 | 7985.0 | 10874.0 | 3856.3 | **26521.6** |
| #Superhuman (↑) | 0 | N/A | 1 | 8 | 10 | 9 | **11** | 10 | **11** | **11** | **11** |
| Mean (↑) | 0.000 | 1.000 | 0.332 | 0.956 | 1.046 | 1.097 | 1.264 | 1.266 | 1.366 | 1.459 | **1.516** |

* These methods are implemented with advanced network backbones such as Transformer and Diffusion for world models.

# 4 EXPERIMENTS

## 4.1 EXPERIMENTAL SETUP

### 4.1.1 BENCHMARKS

We evaluate our approach on two representative POMDP domains: image-based single agent RL and multi-agent RL. The former is evaluated on the well-established Atari 100K benchmark Kaiser et al. (2019), which consists of 26 distinct games, and each agent is only allowed to take 100k actions. To ensure statistical reliability, we conduct all experiments using five independent random seeds, consistent with the evaluation methodology of Alonso et al. (2024). For multi-agent RL, we employ the SMAC benchmark Samvelyan et al. (2019), which features a highly complex setup with multi-agent interactions and includes various adversarial scenarios. Following established practices in multi-agent RL Wen et al. (2022), we evaluate our method on 12 representative scenarios shown in Fig. 3, which span diverse learning difficulties.

## 4.2 RESULTS

### 4.2.1 ATARI

We conduct comprehensive comparisons with strong MBRL methods on the Atari benchmark, including DIAMIND (diffusion-based) Alonso et al. (2024), HarmonyDream Ma et al. (2023), Mu-Dreamer Burchi & Timofte (2024), STORM (transformer-based) Zhang et al. (2023), DreamerV3 Hafner et al. (2023; 2025), IRIS Micheli et al. (2022), TWM Robine et al. (2023), and SimPle Kaiser et al. (2019). As evidenced in Table 1, PO-Dreamer achieves superior overall performance with state-of-the-art mean HNS (Human Normalized Score) of 1.516, surpassing human-level performance in 11 out of 26 games. Notably, our method exhibits remarkable performance in temporally extended scenarios like *ChopperCommand*, *Frostbite*, and *UpNDown* where memory processing is crucial for aiding the agent in decision-making. Especially compared to the baseline DreamerV3, our algorithm shows significant performance improvements in 24 out of 26 scenarios, which demonstrates PO-Dreamer's effectiveness in POMDPs.

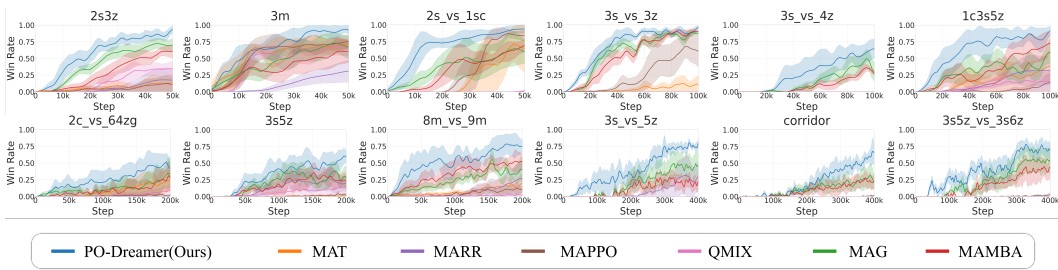

Figure 3: Performance comparisons on SMAC. Solid curves represent the mean of runs over 5 different random seeds, and shaded regions correspond to standard deviation.

### 4.2.2 SMAC

Given the limited number of model-based RL methods for multi-agent settings, we evaluate PO-Dreamer against both established model-free baselines (MAT Wen et al. (2022), MARR Yang et al. (2024), MAPPO Yu et al. (2022) and QMIX Rashid et al. (2020)), as well as MBRL approaches including MAMBA Egorov & Shpilman (2022)) and MAG Wu et al. (2023). As shown in Fig. 3, the training curves exhibit our advantage in both efficiency and effectiveness, which is much more clear in the hard scenarios $3s\_vs\_5z$ and $corridor$. The significant improvement over MAT quantitatively validates the benefits of world models for multi-agent RL. Unlike MAG Wu et al. (2023) and MAMBA Egorov & Shpilman (2022) which require centralized training with global state information, PO-Dreamer achieves superior performance through a decentralized manner. Our superiority indicates Po-Dreamer could handle POMDPs more effectively.

### 4.3 ABLATION STUDY

We conduct ablation studies on 8 representative Atari games, following the set of DIAMOND Alonso et al. (2024). The ablation studies specifically aim to validate 3 parts: (1) the effectiveness of fusing current observations with memory, (2) the contribution of core components including the VAE, action embeddings and positional encoding in the memory encoder, and cross-attention mechanisms in the dynamical predictor, (3) the impact of memory window length $\tau$ and used memory length $\eta$. For detailed analysis of other experimental results (e.g., further discussion about VAE and memory decoder), please refer to the Table 4.

Table 2: The Ablation Study for Fusion Analysis and Component Analysis

| Game | Random | Human | Fusion Analysis | | | | Component Analysis | | | | Ours |
|------|--------|-------|---------|---------|---------|---------|--------|--------|--------|--------|------|
| | | | CE Only | MG Only | CO Only | CO + CO | W/o VAE | W/o CA | W/o Pos | W/o Act | |
| Amidar | 5.8 | 1719.5 | 146.9 | 75.2 | 139.0 | 135.1 | 120.9 | 131.6 | 149.4 | 150.3 | **151.7** |
| Asterix | 210.0 | 8503.3 | **1276.7** | 691.4 | 932.0 | 981.4 | 1074.8 | 1126.3 | 1176.3 | 1255.4 | 1244.3 |
| Breakout | 1.7 | 30.5 | 81.4 | 45.3 | 31.0 | 62.1 | 73.2 | 84.6 | 78.5 | 80.1 | **85.3** |
| Frostbite | 65.2 | 4334.7 | 2006.1 | 1073.8 | 909.0 | 1706.2 | 1952.7 | 1793.5 | 1852.6 | 1936.8 | **2033.7** |
| Hero | 1027.0 | 30826.4 | 10532.0 | 8210.4 | 11161.0 | 11581.7 | 13247.4 | 9644.2 | 11368.2 | 11752.6 | **12258.4** |
| Kangaroo | 52.0 | 3035.0 | 5519.6 | 1249.5 | 4098.0 | 4716.5 | 4825.3 | 5139.7 | 5461.3 | **5637.2** | 5573.9 |
| Krull | 1598.0 | 2665.5 | 7973.9 | 7533.8 | 7782.0 | 7317.5 | 7421.0 | 7982.3 | 7358.2 | 8152.4 | **8261.6** |
| RoadRunner | 11.5 | 7845.0 | 21036.0 | 9347.1 | 15565.0 | 19376.2 | 19664.3 | 20176.9 | 16417.1 | 21896.3 | **22518.2** |
| Mean (↑) | 0.000 | 1.000 | 1.781 | 1.155 | 1.357 | 1.549 | 1.621 | 1.752 | 1.617 | 1.817 | **1.865** |

### 4.3.1 FUSION OF CURRENT OBSERVATION AND MEMORY

We conduct three key experiments to analyze the memory guided mechanism and our model components: (1) We only use the historical observations from the current episode (CE Only), while not utilizing long-term memory information. (2) Individual module evaluation where we isolate either the observation encoder (CO Only) or memory encoder (MG Only); (3) Fusion validation where we replace memory with duplicated current observations (CO+CO) to validate that the improvements come from the fusion of current observation and memory rather than the complicated model.

As demonstrated in Table 2, "CE Only" shows weaker performance compared to our method, indicating that utilizing long-term memory observations more effectively improves world model perfor-

mance compared to relying solely on short-term historical information. "CO only" and "MG only" are much inferior to the full model, and it indicates that they could be complementary to each other. Even though the memory encoder also contains the current observation, it mainly focuses on extracting relative information from memory, which is different from the current encoder. "CO+CO" performs better than "CO only" but still worse than the full model. It shows the complicated model indeed improves performance, while the advantages of PO-Dreamer are from not only the complicated model but also the fusion mechanism.

### 4.3.2 EFFECTS OF COMPONENTS

To evaluate the individual contributions of core model components in the model, we conduct a comprehensive ablation study. The evaluated components include the VAE, action embeddings and positional encoding in the memory encoder, and cross-attention in the fused dynamics predictor. For each experiment, one component is removed at a time while keeping the rest of the architecture unchanged. The results summarized in Table 2 indicate that all ablated variants consistently underperform the complete model. This demonstrates that each component positively contributes to overall performance.

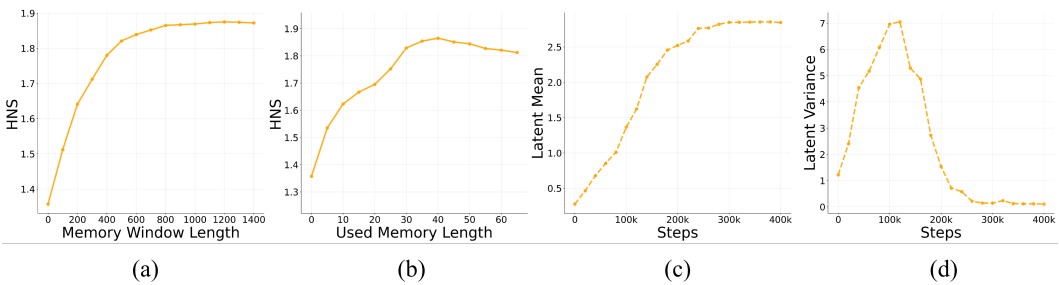

(a)        (b)        (c)        (d)

Figure 4: (a) and (b) are the influence of memory window length $\tau$ and used memory length $\eta$ on HNS performance across eight games. (c) and (d) are the averages of latent mean and latent variance computed across all 20 latent dimensions.

### 4.3.3 PARAMETER ANALYSIS

As shown in Fig. 4(a)(b), the HNS value increases with the memory window length $\tau$, plateauing after $\tau = 800$ and eventually stabilizing. This suggests that a longer memory window enriches the agent's historical information, leading to better decision-making. However, when the used memory length $\eta$ exceeds 40, HNS begins to decline, indicating that while moderate memory enhances performance, excessive information introduces noise and impedes training. To balance performance and efficiency, we set $\tau = 800$ and $\eta = 40$.

To understand the VAE's behavior in the memory encoder, we visualize the mean and variance of the latent variable during learning (Fig. 4(c)(d)). The mean gradually stabilizes and the variance converges to 0, suggesting that the VAE initially promotes exploration of the world model, then shifts toward exploitation as predictions stabilize. This aligns with the exploration–exploitation trade-off in reinforcement learning.

## 5 CONCLUSION

We propose a novel memory guided world model algorithm that extracts critical temporal features from memory sequences, effectively handling the incomplete and noisy state information in POMDPs. Notably, we design a fusion mechanism to integrate current observations with memory, learning a more robust world model. We conduct extensive experiments in both multi-agent and single-agent scenarios, and the results demonstrate that our method exhibits significant advantages in sample efficiency and performance. Building upon the current work, we will explore more advanced network architectures to enhance the modeling capabilities of world models.

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

## A APPENDIX

### A.1 PO-DREAMER ALGORITHM

PO-Dreamer is a world model-based reinforcement learning algorithm that incorporates a memory guided dual-branch state space model. The algorithm consists of two main components: world model learning and reinforcement learning.

---

**Algorithm 1:** Pseudocode for PO-Dreamer

---

1   **Initialize:** World model parameters $\phi$, Reinforcement learning parameters $\theta$, Replay buffer $\mathcal{B}$.
2   **while** *not converged* **do**
3     // Collect experience
4     $\{(o_t, a_t, r_t, c_t, m_t)_{t=1}^T\} \sim \text{Sample}(\mathcal{B})$
5     // World model learning
6     **for** *update step c = 1 ... C* **do**
7       // Initialize the hidden state
8       $h_0^{co}, h_0^{mg} \leftarrow \text{Initialize}()$
9       **for** *time step t = 1 ... T* **do**
10         // Observation Encoder
11         $z_t^{co} \sim q_\phi(z_t^{co}|h_t^{co}, o_t)$
12         // Memory Encoder
13         $z_t^{mg} \sim q_\phi(z_t^{mg}|h_t^{mg}, m_t)$
14         // Fused Sequence model
15         $h_t^{co}, h_t^{mg} = f_\phi(h_{t-1}^{co}, z_{t-1}^{co}, h_{t-1}^{mg}, z_{t-1}^{mg}, a_{t-1})$
16         // Fused Dynamics predictor
17         $\hat{z}_t^{co}, \hat{z}_t^{mg} \sim p_\phi(\hat{z}_t^{co}, \hat{z}_t^{mg} \mid h_t^{co}, h_t^{mg})$
18         // Compute latent state
19         $s_t = \text{CAT}(h_t^m, z_t^m, h_t^p, z_t^p)$
20         // Update world model
21         Compute the total loss $\mathcal{L}(\phi)$ using Eq. 13
22         Update $\phi$
23       **end**
24       // Behavior learning
25       Roll-out within the imagination $\{\hat{s}_t\}_{t=i}^{i+l}$
26       **for** *time step j = i ... i + l* **do**
27         // Imagine an action
28         $a_t \sim \pi_\theta(a_t \mid \hat{s}_t)$
29         // Imagine in world model
30         $\hat{s}_{t+1} \sim p_\phi(\hat{s}_{t+1} \mid \hat{s}_t, a_t)$
31         Update the policy and value models in Eq. 14 using estimated rewards and values.
32       **end**
33     **end**
34     // Environment interaction
35     $s_0 \leftarrow \text{Initialize}()$
36     **for** *time step t = 1 ... T* **do**
37       Sample $a_t \sim \pi_\theta(a_t \mid \hat{s}_{t-1})$
38       $s_t, o_t, r_t, c_t \leftarrow \text{env.step}(a_t)$
39     **end**
40     $\mathcal{B} \leftarrow \mathcal{B} \cup \{(o_t, a_t, r_t, c_t)_{t=1}^T\}$
41 **end**

---

### A.2 IMPLEMENTATION DETAILS

#### A.2.1 NETWORK DESIGN

Since the implementation of the current encoder and the decoders are the same as the set in DreamerV3 Hafner et al. (2023; 2025), here we mainly describe the memory encoder, fused sequence model, fused dynamics predictor and history decoder.

**Memory Encoder** is used to encode the current observation paired with a memory window $\tau = 800$ time steps are encoded through $f_o(\cdot)$ separately to generate $f_o(o_t)$ and a sequence of observation embedding $\{f_o(o_{\hat{t}})\}_{\hat{t}=1}^{\tau}$. The observation embedding is a four-layer CNN, characterized by kernel= 4, stride $= 2$, and padding $= 1$, the output channels of each layer is $\{(32 \times 32 \times 32), (64 \times 16 \times 16), (128 \times 8 \times 8), (256 \times 4 \times 4)\}$. Every layer is combined with a batch normalization layer and a ReLU activation. As the input image $o_t$ is $3 \times 64 \times 64$, it is transformed into $256 \times 4 \times 4$ by the CNN. It is then flattened into a 4096-dimensional vector and then mapped through a linear layer into a 512-dimensional $f_o(o_t)$ observation embeddings. And then the adaptive dropout mechanism is used to select the most relevant memory fragments by computing the cosine similarity $S$ between memory embeddings and the current observation embedding to select top-$\eta$ memory, we set the selected memory length $\eta = 40$. The action embedding is a two-layer MLP, combined with a batch normalization layer and a ReLU activation. The action vector is mapped into a 512-dimensional $f_a(a_{\hat{t}})$ action embeddings. In addition, the positional encoding of memory embeddings $\hat{t} \in \{1, \eta\}$ is computed as Vaswani et al. (2017), denoted as $\{Pos_{\hat{t}}\}_{\hat{t}=1}^{\eta}$ with the dimension of 512. Then, we construct the context embedding through element-wise addition of observation embedding, action embeddings, and positional encoding as $f_{\hat{t}}^c = f_o(o_{\hat{t}}) + f_a(a_{\hat{t}}) + Pos_{\hat{t}}$ where $\hat{t} \in (1, \eta)$, and the context embedding of current observation is set as $f_t^c = f_o(o_t)$. That, is $f_t^c \in \mathbb{R}^{1 \times 512}$, and $F^c \in \mathbb{R}^{\eta \times 512}$ which concertante $\{f_{\hat{t}}^c\}_{\hat{t}=1}^{\eta}$ together. The Multi-head Attention (MHA) Vaswani et al. (2017) is used in Eq 5. In the transformer blocks, the number of blocks is 3 and transformer head is 8 with the hidden dimension is 512. And then we obtain $f_t$ with the dimension of 512. Then it is encoded by the VAE Kingma et al. (2013) block, which contains a two-layer MLP with BN and ReLU, the first layer maps $f_t$ into 256. And then it is mapped to the 20-dimensional mean $\mu_\phi(f_t)$ and standard deviation $\sigma_\phi(f_t)$ by two MLPs respectively.

**Fused sequence model** models the recurrent state sequence in the world model, the sequence model is implemented as two independent GRUs respectively. The input is the concatenation of latent state and action $\text{CAT}(h_t^{co}, z_t^{co}, h_t^{ha}, z_t^{co}, a_t)$. The output dimensions of both GRU networks are 512.

**Fused dynamics predictor** learns to predict the $\hat{z}_t^{co}$ and $\hat{z}_t^{ha}$ according to the Eq. 3. There is a cross-attention mechanism block to fuse the outputs of two GRUs. The cross-attention is an 8-heads MHA Vaswani et al. (2017) with 512-dimension hidden feature. Then the outputs $h_t^{co'}, h_t^{ha'}$ are mapped to $\hat{z}_t^{co}, \hat{z}_t^{ha}$, which follows the set of DreamerV3 Hafner et al. (2025), and the dimension of outputs $\hat{z}_t^{co}, \hat{z}_t^{ha}$ is 1024.

**Memory decoder** is designed to reconstruct memory information $f_t$, which is a 3-layer MLPs (each with BN and ReLU) to map the latent state $s_t \doteq \text{CAT}(h_t^{co}, z_t^{co}, h_t^{ha}, z_t^{co})$ into the memory $\hat{f}_t$, where the dimension of $s_t$ is 3072 and the dimension of $\hat{f}_t$ is 512.

### A.2.2 HYPERPARAMETERS

### A.2.3 IMPLEMENTATIONS DETAILS

Table 3: Hyperparameters. This configuration aligns with the setup used in DreamerV3 Hafner et al. (2023; 2025).

| Name | Symbol | Value |
|---|---|---|
| Replay capacity | — | $5 \times 10^6$ |
| Batch size | $B$ | 16 |
| Batch length | $T$ | 64 |
| Imaginationhorizon | $L$ | 16 |
| Learning rate | — | $4 \times 10^{-5}$ |
| Optimizer | — | Adam |
| Imagination horizon | $H$ | 15 |
| Dropout probability | $p$ | 0.1 |
| Representation loss scale | $\beta_{\text{rep}}$ | 0.1 |
| Dynamics loss scale | $\beta_{\text{dyn}}$ | 0.5 |
| Return lambda | $\lambda$ | 0.95 |
| Memory window length | $\tau$ | 800 |
| Selected memory length | $\eta$ | 40 |

Building upon the DreamerV3 framework, our method maintains the original architecture and training hyperparameters unless specified in previous sections. The model processes environmental inputs using specialized modules: a 5-layer CNN for Atari's visual observations and a 5-layer MLP for StarCraft's vector data. We set the memory window length $\tau = 800$ and the used length $\eta = 40$ timesteps across all experiments and employ the Adam optimizer. We conduct independent training runs per environment using 5 distinct random seeds, which is the same as DIAMOND Alonso et al. (2024). The details of the used hyperparameters refer to Table 3.

### A.3 MORE COMPONENT ANALYSIS

Here we conduct two additional ablation studies to validate: (1) the role of the VAE in modeling the stochasticity of unobservable components, and (2) the effectiveness of the memory decoder.

The results are summarized in Table 4. Since the contribution of the VAE in the memory encoder has already been validated in the "W/o VAE" experiment, we here introduce the VAE into the current observation encoder, denoted as the "CO + VAE". The results show that adding the VAE to the current encoder degrades performance. This suggests that a naive application of the VAE is ineffective, and that incorporating it into the memory encoder better aligns with the characteristics of partially observable environments. Furthermore, when the memory decoder is removed ("W/o MD"), performance drops compared to the full model, confirming the positive contribution of the memory decoder. This is because the current observation decoder alone cannot fully represent the complete state.

Table 4: Additional ablation study results, where MD is the memory decoder.

| Game | CO + VAE | W/o MD | Ours |
|---|---|---|---|
| Amidar | 133.5 | 143.1 | **151.7** |
| Asterix | 913.1 | 1114.2 | **1244.3** |
| Breakout | 41 | 73.3 | **85.3** |
| Frostbite | 1270.8 | 1796.8 | **2033.7** |
| Hero | 10774.3 | 11704.1 | **12258.4** |
| Kangaroo | 4778.3 | 5297.3 | **5573.9** |
| Krull | 6921 | 7961.4 | **8261.6** |
| RoadRunner | 16804.3 | 19468.2 | **22518.2** |
| Mean (↑) | 1.355 | 1.752 | **1.865** |

### A.4 QUANTITATIVE ANALYSIS COMPARISON WITH DREAMERV3

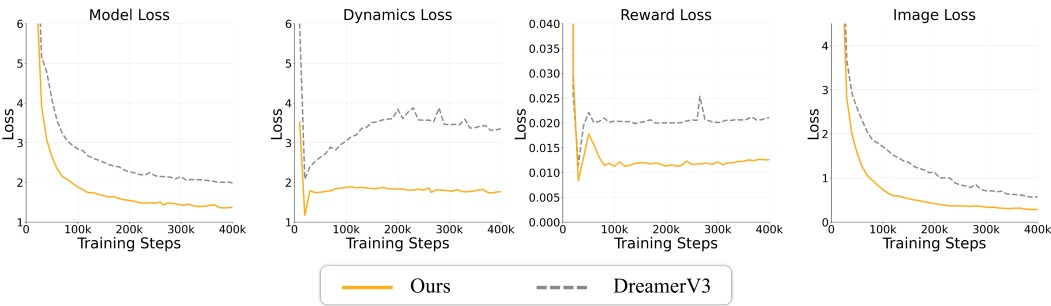

Figure 5: The curves of the averaged losses (including model loss, dynamics loss, reward loss, and image loss) across eight games in the Atari benchmark.

For a precise evaluation of world model accuracy, we compare the training loss curves of our algorithm with DreamerV3Hafner et al. (2023; 2025). The comparison is based on experiments conducted on eight Atari games, with each game run five times using different random seeds to ensure reliable and consistent results. The average loss curves are presented in Fig. 5. A direct comparison shows that our method converges more rapidly and to a lower asymptotic value on every loss component—namely the model (Eq. 13), reward, image (Eq. 11), and dynamics losses (Eq. 12). The

consistent superiority across these metrics underscores that our approach learns a more accurate and robust world model, which is critical for effective decision-making in POMDPs.

## A.5 QUALITATIVE VISUAL COMPARISON WITH DREAMERV3

We now compare to DreamerV3 Hafner et al. (2023; 2025),

We further complement the quantitative results with a qualitative visualization to more directly compare the world model accuracy of our method versus DreamerV3 Hafner et al. (2023; 2025). To isolate modeling capability from training dynamics and ensure an equitable comparison, we trained both models on a shared, static dataset comprising 100k frames collected by an expert policy. The subsequent analysis focuses on the accuracy of their world model reconstructions.

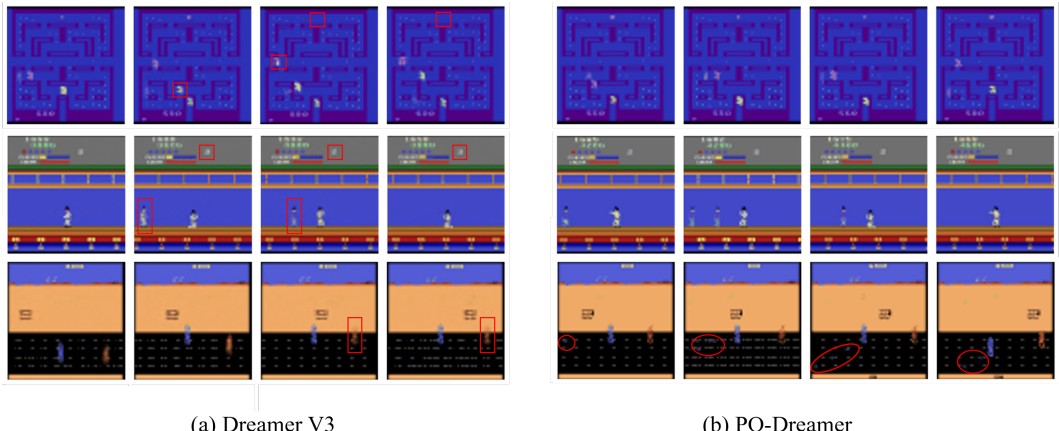

(a) Dreamer V3           (b) PO-Dreamer

Figure 6: Frames imagined by (a) Dreamer V3 and (b) PO-Dreamer. Red boxes highlight inconsistencies that occur only in trajectories generated by Dreamer V3. In Alien (top row), Dreamer V3 exhibits rendering artifacts, including incorrect enemy coloring and a missing key game element (the gem), while PO-Dreamer produces a faithful reconstruction. In KungFuMaster (middle row), Dreamer V3 fails to accurately estimate the player's remaining lives and misidentifies the enemy type, whereas PO-Dreamer correctly captures both. In Road Runner (bottom row), Dreamer V3 omits environmental rewards (small blue dots on the road), while PO-Dreamer reconstructs them precisely (red circle) and generates a more accurate background reconstruction provides crucial contextual information, allowing for effective differentiation between the player character and the background. Additionally, our method shows some improvement in reconstructing image details compared to Dreamer V3.

As shown in Figure 6 that the trajectories imagined by PO-Dreamer are generally of higher visual quality and more faithful to the true environment compared to the trajectories imagined by Dreamer V3. In particular, the trajectories generated by Dreamer V3 exhibit visual inconsistencies and state discontinuities across frames, leading to the misrepresentation of critical game elements such as enemies or rewards. Although these artifacts may occupy only a few pixels in the generated images, they can substantially impact the reinforcement learning process. For instance, in the Alien environment, rendering errors in enemies and missing gem objects may misguide the agent's avoidance behavior and reward acquisition, thereby affecting policy learning. Improvements in the accuracy of the world model on such visual details contribute to more stable and effective policy optimization.

## A.6 ANONYMOUS CODE REPOSITORY

To ensure reproducibility and foster further research, the source code and project website for our work are provided in the following anonymous repository:

```
https://anonymous.4open.science/r/PO-Dreamer_anonymous
```

# B ADDITIONAL PARTS

## B.1 ABLATION STUDY ON MEMORYMAZE

The ablation experiments on MemoryMaze (from 9×9 to 15×15) confirm that PO-Dreamer's superiority in handling partial observability and long-horizon navigation stems from its core fusion mechanism and complementary key components. Fusion analysis shows that single-branch variants (MG only/CO only) and current-episode-only memory (CE Only) underperform across maze sizes—especially on large 13×13-15×15 mazes—proving that integrating cross-episode long-term memory with current observations is critical for retaining path information. The CO+CO variant (duplicated current observation encoding) also lags behind, ruling out model capacity expansion as the primary gain driver. Component analysis further demonstrates that omitting any core module (VAE, cross-attention, positional encoding, action embeddings) degrades performance: VAE enhances robustness against noise, cross-attention enables bidirectional feature interaction, positional encoding preserves temporal order, and action embeddings link past actions to observations—all collectively supporting reliable navigation. PO-Dreamer achieves the best performance on 11×11, 13×13, and 15×15 mazes (and second-best on 9×9), validating that its integrated design effectively addresses the challenges of partially observable POMDPs.

Table 5: Ablation study on memorymaze

| Game | Fusion Analysis | | | | Component Analysis | | | | Ours |
|---|---|---|---|---|---|---|---|---|---|
| | CE Only | MG | CO | CO + CO | W/o VAE | W/o CA | W/o Pos | W/o Act | |
| 9 × 9 | 37.5±1.4 | 31.4±0.7 | 35.9±0.4 | 36.1±0.8 | 34.2±0.9 | 33.1±0.5 | 36.1±1.3 | **38.4±1.8** | 38.0±1.6 |
| 11 × 11 | 46.1±1.3 | 32.6±1.4 | 43.5±1.1 | 45.2±1.7 | 33.5±1.5 | 36.5±1.2 | 44.8±1.5 | 45.6±1.7 | **47.6±1.5** |
| 13 × 13 | 48.3±1.7 | 37.5±1.5 | 36.2±1.2 | 31.5±1.9 | 37.9±1.7 | 40.2±1.5 | 49.1±2.3 | 47.2±2.1 | **50.3±2.2** |
| 15 × 15 | 43.0±2.5 | 39.4±2.3 | 15.7±1.3 | 16.5±2.2 | 34.2±2.1 | 37.0±2.0 | 45.1±2.1 | 47.1±2.5 | **47.4±2.7** |

*Note:* The **bold values** indicate the best performance among all, while the underlined values indicate the second-best performance.

## B.2 COMPARISON WITH DREAMERV3 ON SMAC

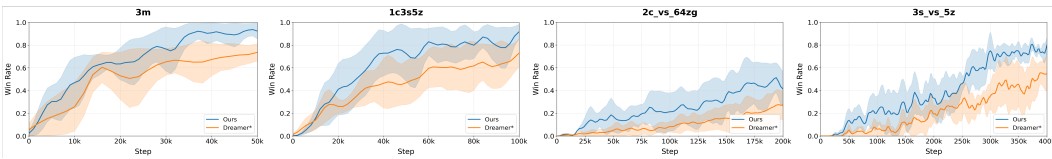

Figure 7: Performance comparison between the PO-Dreamer and the modified DreamerV3 algorithm (denoted as Dreamer*) across scenarios with varying difficulty levels.

We selected four scenarios with varying difficulty levels to compare the performance of our proposed algorithm against our modified DreamerV3 algorithm (denoted as Dreamer*). It can be observed that the PO-Dreamer achieves a remarkable performance improvement compared with the modified DreamerV3 algorithm (Dreamer*).

## B.3 TRAINING CURVE

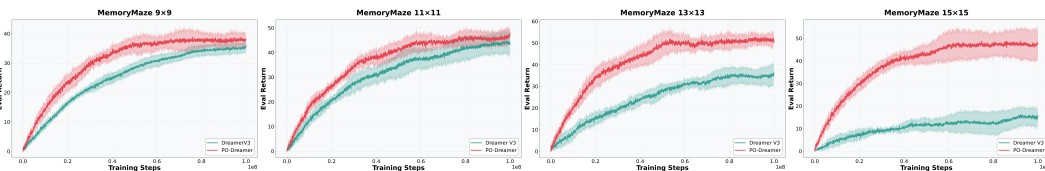

Figure 8: The curves of the return in the MemoryMaze benchmark.

## B.4  Computational Analysis of PO-Dreamer

To address concerns about the computational overhead introduced by PO-Dreamer's cross-attention mechanism (involved in both training and inference for fusing current observations and memory), we provide quantitative metrics for transparency and validation. This design is critical for our model's performance, and we demonstrate that the overhead is well-controlled while justifying the performance gains.

Table 6: Comprehensive Breakdown of Computational Metrics

| Metric | DreamerV3 | DIAMOND | STORM | PO-Dreamer |
|---|---|---|---|---|
| Total Parameters | 14M | 13M | 25M | 27M |
| Parameter | -model_opt: 11.29M -actor_opt: 1.05M -value_opt: 1.18M | -denoiser: 4.41M -rew_end_model: 5.90M -actor_critic_opt: 3.23M | -model_opt: 21.71M -actor_critic_opt: 3.42M | -model_opt: 24.96M -actor_opt: 1.05M -value_opt: 1.18M |
| FLOPs | 6.02 GFLOPs | 29.71 GFLOPs | 4.71 GFLOPs | 13.48 GFLOPs |
| Inference Parameter | 9M | 13M | 25M | 17M |
| Inference Latency (s)* | 8.7±0.3 | 31.7±1.4 | 21.7±0.8 | 12.3±0.4 |
| Mean HNS(Atari) | 1.097 | 1.459 | 1.266 | 1.516 |

Note: Inference latency is measured as the average value obtained by running 100 steps on the NVIDIA A100.

In summary, while PO-Dreamer's introduces computational overhead in design, our quantitative analysis (Table 6) confirms this overhead is tightly constrained via architectural optimizations. The superior performance across benchmarks validates a well-balanced trade-off between complexity and effectiveness, rendering PO-Dreamer both expressive and practically feasible.

## B.5  Future Directions

While PO-Dreamer achieves promising performance in partially observable reinforcement learning tasks, there remains substantial room for optimization in model complexity and training efficiency, which we plan to explore in future work.

First, in terms of network architecture advancement, we will integrate more efficient sequence modeling backbones to replace the current GRU-based fused sequence model and the Transformer-based cross-attention module. Second, regarding architectural efficiency optimization, we aim to design a more compact dual-branch fusion mechanism. Finally, we will also explore transfer learning paradigms, pre-training the memory encoder and world model on general POMDP benchmarks (e.g., Crafter, MemoryMaze) and fine-tuning them on specific tasks, thereby reducing the number of training steps required for convergence. These improvements will further balance the performance and complexity of PO-Dreamer, expanding its applicability to resource-constrained scenarios.

