# OpenReview forum: "PO-Dreamer: Memory Guided World Models for Partially Observable Reinforcement Learning"
_ICLR.cc/2026/Conference — ICLR 2026 Conference Withdrawn Submission_

### Official Review · Reviewer_w5xG · 2025-10-22

**Soundness:** 3
**Presentation:** 2
**Contribution:** 2
**Rating:** 4
**Confidence:** 4

**Summary:**

The paper introduces PO-Dreamer, an extension of the Dreamer algorithm designed to address challenges posed by partial observability. This enhancement incorporates a novel internal memory state intended to emulate the behavior of an external memory, which aggregates past observations from both the current and previous episodes. The central idea is to enable Dreamer to more effectively model non-stationary and partially observable dynamics by leveraging this memory mechanism. Empirical results on the Atari 100k benchmark demonstrate that the inclusion of the memory state boosts the performance of Dreamer-V3. Furthermore, in the multi-agent StarCraft environment, PO-Dreamer surpasses several state-of-the-art approaches.

**Strengths:**

The paper introduces a theoretically sound extension of Dreamer by integrating an external memory module, although I have some concerns regarding the observation selection strategy outlined in Equation 4. An innovation in PO-Dreamer is its inclusion of observations from previous episodes in the memory, which differentiates it from earlier approaches. Empirical results show that PO-Dreamer achieves strong performance on both the Atari 100k and SMAC benchmarks, surpassing competitive baselines such as DreamerV3 in the Atari domain.

**Weaknesses:**

It is unclear why the proposed method works. The paper asserts that augmenting Dreamer with external memory improves its ability to handle partial observability. However, the environments used to evaluate PO-Dreamer are generally regarded as fully observable in the existing literature. Partial observability typically poses a challenge in reinforcement learning when tasks involve long-term dependencies that require the agent to retain information over time to act optimally. For example, Hausknecht & Stone (2015) introduced artificial occlusions—such as random frame dropping—to simulate partial observability in Atari games. Another well-known case is the T-maze (Esslinger et al., 2022, Fig. 3a), where the agent receives a clue at the beginning of the episode that it must remember to make the correct decision later. As far as I can tell, all the works cited in Section 2.2 deliberately evaluate their PO agents in settings where memory is essential for task completion. This does not appear to be the case for PO-Dreamer. Consequently, I find the evidence insufficient to support the paper’s first claimed contribution.

> "A memory guided world model, PO-Dreamer, is proposed to extract key temporal information to infer unobservable aspects of the environment, effectively addressing the challenges of non-stationarity in POMDPs"

I find it difficult to see how the experiments substantiate the claim that PO-Dreamer effectively extracts key temporal information to infer unobservable aspects of the environment. This concern arises primarily because the method was not evaluated on environments where temporal information is critical for solving the task—beyond simply recalling the last few frames, as is typical in Atari. I believe this also largely applies to the SMAC domains used in the evaluation.

Furthermore, the mechanism used to populate the external memory (Equation 4) appears to contradict the goal of retaining important past information to address partial observability. If I understood correctly, the memory is populated with observations that are similar to the current one, based on cosine similarity between their embedding vectors. In genuinely partially observable settings, where success depends on recalling information from the distant past, retrieving observations similar to the current one is not helpful. Such observations are likely redundant, as they reflect what the agent is already seeing. What is needed instead is access to information that is currently missing but crucial for decision-making—such as the initial clue in the T-maze task.

Additionally, the decision to include observations from previous episodes seems misaligned with the goal of uncovering unobservable aspects of a POMDP. Consider again the T-maze: the clue changes in every episode. Therefore, retrieving observations from past episodes could mislead the agent, as it might attend to outdated or irrelevant clues, rather than the one pertinent to the current episode.

In summary, while the paper presents strong evidence that PO-Dreamer performs well in fully observable domains, I am not convinced that its success is attributable to improved handling of partial observability. It seems more plausible that the gains stem from increased model capacity or richer learning signals, rather than from mechanisms that effectively address partial observability.

**Questions:**

1. What accounts for PO-Dreamer's superior performance over DreamerV3 in the Atari 100k benchmark? Given the concerns raised earlier regarding partial observability, it seems unlikely that the observed performance gains are primarily due to improved inference of unobserved aspects of the environment.

2. What is the rationale for including observations from different episodes in the external memory? In tasks where episode-specific information is critical (e.g., the T-maze), retrieving observations from other episodes could introduce noise or confusion rather than aid decision-making.

3. What is the motivation behind the observation selection strategy described in Equation 4? Specifically, why is similarity to the current observation—measured via cosine similarity between embeddings—a meaningful criterion? Would alternative strategies, such as random sampling or task-specific heuristics, perform better in partially observable settings?

4. How does PO-Dreamer perform in environments that are explicitly designed to test partial observability? For example, how does it compare to prior methods on benchmarks used in the works cited in Section 2.2, where memory is essential for task success?

---

> ### Author Response · Authors · 2025-11-23
> **Response to AnonReviewer4(Reviewer w5xG)**
>
> We sincerely appreciate your review!
>
> 1.	Reasons for PO-Dreamer’s Superior Performance Over DreamerV3 on Atari 100K
>
> The performance gain of PO-Dreamer over DreamerV3 on the Atari 100K benchmark primarily stems from the improved accuracy of the world model in modeling partially observable dynamics, rather than trivial factors such as increased model capacity. We validate this conclusion through the following analysis: As shown in Appendix A4 of the revised manuscript, PO-Dreamer achieves a much lower state reconstruction loss and a lower reward prediction error than DreamerV3. This confirms that our memory-guided world model captures environmental dynamics more precisely than DreamerV3.
>
> 2.	Rationale for Including Cross-Episode Observations in External Memory
>
> The inclusion of cross-episode observations is designed to avoid noise in episode-specific tasks. For tasks with strong episode-specificity, PO-Dreamer’s adaptive dropout mechanism automatically down-weights cross-episode observations with low cosine similarity to the current episode’s key features. For most Atari and SMAC tasks (where environmental rules and enemy strategies are consistent across episodes), cross-episode memory enables the model to learn reusable patterns (e.g., enemy formation strategies in SMAC).
>
> 3.	Motivation for Cosine Similarity-Based Observation Selection (Equation 4)
>
> The cosine similarity criterion in Equation 4 is chosen for its ability to capture semantic relevance between current and historical observations, which is critical for partial observability.
>
> 4.	Performance of PO-Dreamer on POMDP Benchmarks for Memory Evaluation
>
> To address your concern about experimental validity in partial observability, we conducted additional experiments on three canonical POMDP benchmarks where memory is essential. (Refer to Revised Response to Reviewers)
> Thanks sincerely.

---

> ### Author Response · Authors · 2025-11-26
>
> Thanks for your valuable comments on the partial observability validation of our work. We would like to clarify the partial observability of the experimental environments and supplement the generalization experiments as follows:
>
> 1. Explanation of Partial Observability in Experimental Environments
> Regarding the partial observability of Atari environments, we align with the viewpoint of DRQN [1]: *"Real-world tasks often feature incomplete and noisy state information resulting from partial observability like that found in POMDPs. As Figure 1 shows, given only a single game screen, many Atari 2600 games are POMDPs. One example is the game of Pong in which the current screen only reveals the location of the paddles and the ball, but not the velocity of the ball. Knowing the direction of travel of the ball is a crucial component for determining the best paddle location."* Even with frame stacking, the dynamic state information (e.g., object velocity, hidden state transitions) makes Atari games non-fully observable scenarios.
>
> For multi-agent environments (SMAC), partial observability is manifested in two core aspects: (1) Each agent has a limited observation range and can only perceive local information rather than the global state; (2) The mutual interaction between agents leads to non-stationary environmental dynamics, where the action of one agent changes the state space perceived by others.
>
> 2. Supplementary Generalization Experiments on Specialized POMDP Benchmarks
> To verify the generalization of PO-Dreamer in more challenging partial observable scenarios, we conducted additional experiments on **Crafter** [2] and **MemoryMaze** [3]—two benchmarks specifically designed for testing memory and partial observability capabilities. The experimental results are presented below:
>
> 2.1 Crafter Experiment Results
> Crafter requires agents to collect resources, build tools, and complete complex tasks in an open world, where long-term memory of resource locations and task progress is essential.
>
> | Method       | Score       | Reward      |
> |--------------|-------------|-------------|
> | Human        | 50.5±6.8    | 14.3±2.3    |
> | PPO          | 4.6±0.3     | 4.2±1.2     |
> | Rainbow      | 4.3±0.2     | 6.0±1.3     |
> | DreamerV3    | 14.5±1.6    | 11.7±1.9    |
> | Ours (PO-Dreamer) | 16.2±1.3 | 12.3±1.4 |
>
> 2.2 MemoryMaze Experiment Results
> MemoryMaze is a 3D maze environment where agents need to remember the location of target points and navigation paths to complete tasks. We tested mazes of different sizes to evaluate long-term memory capabilities.
>
> | Method       | Memory9x9   | Memory11x11 | Memory13x13 | Memory15x15 |
> |--------------|-------------|-------------|-------------|-------------|
> | Human        | 26.4±1.6    | 44.3±2.5    | 55.5±3.5    | 67.7±4.0    |
> | IMPALA       | 23.4±0.4    | 28.2±0.3    | 27.5±0.6    | 17.7±1.4    |
> | DreamerV3    | 35.9±0.4    | 43.5±1.1    | 36.2±1.2    | 15.7±1.3    |
> | Ours (PO-Dreamer) | 38.0±1.6 | 47.6±1.5 | 50.3±2.2 | 47.7±2.7 |
>
> As shown in the results, PO-Dreamer outperforms baseline methods (including DreamerV3) on both benchmarks, especially in large-scale MemoryMaze (15x15) where long-term memory is critical—PO-Dreamer maintains stable performance, while DreamerV3 and IMPALA experience significant performance degradation. This confirms the effectiveness of our memory mechanism in handling complex partial observable scenarios.
>
>
>
>
> **References**
> [1] Hausknecht M, Stone P. Deep Recurrent Q-Learning for Partially Observable MDPs[C]//AAAI Symposia, 2015.
> [2] Hafner D. Benchmarking the spectrum of agent capabilities[J]. arXiv preprint arXiv:2109.06780, 2021.
> [3] Pasukonis J, Lillicrap T, Hafner D. Evaluating long-term memory in 3d mazes[J]. arXiv preprint arXiv:2210.13383, 2022.

---

### Official Review · Reviewer_Cf3s · 2025-10-23

**Soundness:** 2
**Presentation:** 2
**Contribution:** 2
**Rating:** 2
**Confidence:** 4

**Summary:**

The paper introduces PO-Dreamer, an extension of Dreamer that incorporates a memory module to improve performance in partially observable environments. The added memory stores a subset of past observations and actions, which may come from either the current or previous episodes. During interaction, the model selectively retains a smaller subset of this memory and performs memory lookup using a Transformer, with the current observation serving as the query. The retrieved memory is then fused to augment the latent state in Dreamer. The approach is evaluated on single-agent Atari 100K and multi-agent SMAC benchmarks, showing improvements on some tasks compared to selected baselines.

**Strengths:**

- The core problem addressed in the paper, namely the memory limitations in model-based reinforcement learning, is both relevant and significant.
- The concept of leveraging cross-episode data during inference is novel.
- Overall, the paper is well organized and easy to follow.
- Ablation studies are included and help clarify the contribution of different components of the method.

**Weaknesses:**

- The main motivation of the paper comes from the partial observability of the environment. However, in the selected single-agent tasks, the observation function is only mildly partial with deterministic transitions, limiting evidence for the method’s effectiveness in harder scenarios.

- While the experiments demonstrate some improvement over existing baselines, they do not provide clear evidence of stronger memory capabilities. To support this claim, the authors could consider long-horizon tasks that require the agent to retain information from the distant past.

- No error bars or statistical significance are reported for Atari 100K, making results less reliable.


- Table 2 suggests that increased model capacity (CO+CO) improves performance compared to Dreamer. This raises the question of how Dreamer with a larger latent state size would perform and whether the observed gains are primarily due to increased model capacity rather than the method itself.

- The proposed method requires storing raw observations in memory and processing the entire memory at each timestep, which introduces additional time and memory complexity. This overhead should be analyzed and discussed in depth in the paper.

- Although the authors claim to address non-stationarity in POMDPs, both the transition function and rewards are assumed to be stationary in the proposed method and in the experimental environments.

- The notations are somewhat overcomplicated, and some equations require minor adjustments (e.g., indexing in Eq. 4).

- Some important detail of the method are missing. Please see the questions.

**Questions:**

- Why do you encode timesteps in memory embeddings? My intuition is that the improvement observed from adding timesteps to the memory embeddings might be task-specific. If that’s the case, comparing these results with baselines that don’t use timesteps could be unfair (though I’m not sure whether all of them exclude timesteps).

- What is the trade-off being referred to in line 193?

- Do you stop gradients when computing (4)?

- What is the intuition behind using cross-attention in (8)?

- How do you fill memory in training and inference?

- Why is DreamerV3 missing in Figure 3?

- What is the connection between the exploration–exploitation discussion in line 474 and Figure 4 (c)–(d)?

**Details Of Ethics Concerns:**

There are some unusual hyperref usages in the paper. For example, on page 2, the page number is linked; on page 3, the "Under review as a conference paper at ICLR 2026" header and an invisible rectangle on the left side of the page also appear as hyperlinks.

---

> ### Author Response · Authors · 2025-11-23
> **Response to AnonReviewer3(Reviewer Cf3s)**
>
> We sincerely appreciate your careful review. Below is a detailed response to your inquiries and concerns:
>
> #### 1. Clarification on Positional Encoding in Memory Embeddings
> Thank you for your question about time-step encoding in memory embeddings—we clarify that we **do not encode explicit time steps**; instead, we incorporate **positional encoding** during Transformer encoding. The core purpose of positional encoding is to distinguish the sequential order of observation and action sequences in the memory buffer, rather than marking absolute time steps. This design ensures the model can capture the temporal correlation of historical data (e.g., the order of "collecting a key → approaching a door" in Atari games) without relying on task-specific time-step features.
>
> #### 2. Explanation of the "Trade-off" in Line 193
> The "trade-off" mentioned in Line 193 refers to the **balance between utilizing current observation information and historical memory information** in the memory fusion module. Specifically:
> - On one hand, the model needs to extract high-level features from current observations to capture real-time environmental states;
> - On the other hand, it must retrieve useful historical memory features (e.g., long-term dependency information such as enemy formation in SMAC) that contribute to state reconstruction and dynamics prediction.
>
> The cross-attention mechanism in our method dynamically adjusts the attention weights of current and historical information to achieve this balance, ensuring neither type of information is overemphasized or ignored.
>
> #### 3. Gradient Handling in Equation (4)
> We confirm that **no gradient clipping (or truncation) is applied when calculating Equation (4)**.
>
> #### 4. Intuitive Motivation for Cross-Attention in Equation (8)
> The cross-attention mechanism in Equation (8) is designed to address the **non-independence of information between the two branches (current observation and historical memory)**.
>
> #### 5. Memory Population in Training and Inference Phases
> The memory buffer is populated differently in the training and inference phases, as detailed below:
> - **Training phase**: We first use a random policy to interact with the environment and collect agent trajectory data (including observations, actions, rewards, and termination signals). These trajectories are stored in a replay buffer and used to populate the memory module, providing historical data for memory encoding and fusion.
> - **Inference phase**: The memory module leverages the trajectory data stored in the replay buffer during training (rather than real-time collected data) to retrieve historical information relevant to the current observation, ensuring stable decision-making without additional environmental interaction overhead.
>
> #### 6. Reason for the Absence of DreamerV3 in Figure 3
> DreamerV3 is a **single-agent reinforcement learning algorithm** and cannot be directly applied to the multi-agent SMAC environment (the focus of Figure 3) without significant modification (e.g., adding multi-agent coordination mechanisms). To ensure fair comparison, we only included multi-agent baselines (e.g., MAPPO, QMIX, MAMBA) in Figure 3.
>
> #### 7. Connection Between Exploration-Exploitation Discussion (Line 474) and Figure 4(c)-(d)
> The discussion in Line 474 is closely linked to the results in Figure 4(c)-(d), which depict the trends of the VAE network’s mean and standard deviation of latent states during training:
> - **Mean trend**: The mean of the VAE latent states gradually stabilizes, indicating that the model’s learned latent state space of the environment becomes consistent and reliable.
> - **Standard deviation trend**: The standard deviation approaches zero, reflecting that the model’s uncertainty about environmental states decreases as training progresses.
>
> These trends align with the exploration-exploitation dynamics of reinforcement learning: in the early training stage, the policy tends to explore a larger action space (corresponding to high variance in latent states), while in the later stage, the policy stabilizes and leans toward exploiting historical experience (corresponding to low variance in latent states). We have added a detailed interpretation of this connection in the caption of Figure 4 to strengthen the logical link between the discussion and the results.
>
> #### 8. Correction of Unconventional Hyperlink Usage (Ethics Review Concern)
> We apologize for the unconventional hyperlink usage in the manuscript, which stems from **template-related issues**: the hyperlinks (e.g., page numbers on Page 2, the header on Page 3, and the invisible rectangle) are automatically generated when citations span two pages, and all hyperlinks point to the reference section of the paper (no external or irrelevant links).
>
> Thanks!

---

> ### Author Response · Authors · 2025-11-26
>
> Thanks for your valuable comments on the partial observability validation of our work. We would like to clarify the partial observability of the experimental environments and supplement the generalization experiments as follows:
>
> 1. Explanation of Partial Observability in Experimental Environments
> Regarding the partial observability of Atari environments, we align with the viewpoint of DRQN [1]: *"Real-world tasks often feature incomplete and noisy state information resulting from partial observability like that found in POMDPs. As Figure 1 shows, given only a single game screen, many Atari 2600 games are POMDPs. One example is the game of Pong in which the current screen only reveals the location of the paddles and the ball, but not the velocity of the ball. Knowing the direction of travel of the ball is a crucial component for determining the best paddle location."* Even with frame stacking, the dynamic state information (e.g., object velocity, hidden state transitions) makes Atari games non-fully observable scenarios.
>
> For multi-agent environments (SMAC), partial observability is manifested in two core aspects: (1) Each agent has a limited observation range and can only perceive local information rather than the global state; (2) The mutual interaction between agents leads to non-stationary environmental dynamics, where the action of one agent changes the state space perceived by others.
>
> 2. Supplementary Generalization Experiments on Specialized POMDP Benchmarks
> To verify the generalization of PO-Dreamer in more challenging partial observable scenarios, we conducted additional experiments on **Crafter** [2] and **MemoryMaze** [3]—two benchmarks specifically designed for testing memory and partial observability capabilities. The experimental results are presented below:
>
> 2.1 Crafter Experiment Results
> Crafter requires agents to collect resources, build tools, and complete complex tasks in an open world, where long-term memory of resource locations and task progress is essential.
>
> | Method       | Score       | Reward      |
> |--------------|-------------|-------------|
> | Human        | 50.5±6.8    | 14.3±2.3    |
> | PPO          | 4.6±0.3     | 4.2±1.2     |
> | Rainbow      | 4.3±0.2     | 6.0±1.3     |
> | DreamerV3    | 14.5±1.6    | 11.7±1.9    |
> | Ours (PO-Dreamer) | 16.2±1.3 | 12.3±1.4 |
>
> 2.2 MemoryMaze Experiment Results
> MemoryMaze is a 3D maze environment where agents need to remember the location of target points and navigation paths to complete tasks. We tested mazes of different sizes to evaluate long-term memory capabilities.
>
> | Method       | Memory9x9   | Memory11x11 | Memory13x13 | Memory15x15 |
> |--------------|-------------|-------------|-------------|-------------|
> | Human        | 26.4±1.6    | 44.3±2.5    | 55.5±3.5    | 67.7±4.0    |
> | IMPALA       | 23.4±0.4    | 28.2±0.3    | 27.5±0.6    | 17.7±1.4    |
> | DreamerV3    | 35.9±0.4    | 43.5±1.1    | 36.2±1.2    | 15.7±1.3    |
> | Ours (PO-Dreamer) | 38.0±1.6 | 47.6±1.5 | 50.3±2.2 | 47.7±2.7 |
>
> As shown in the results, PO-Dreamer outperforms baseline methods (including DreamerV3) on both benchmarks, especially in large-scale MemoryMaze (15x15) where long-term memory is critical—PO-Dreamer maintains stable performance, while DreamerV3 and IMPALA experience significant performance degradation. This confirms the effectiveness of our memory mechanism in handling complex partial observable scenarios.
>
>
>
>
> **References**
> [1] Hausknecht M, Stone P. Deep Recurrent Q-Learning for Partially Observable MDPs[C]//AAAI Symposia, 2015.
> [2] Hafner D. Benchmarking the spectrum of agent capabilities[J]. arXiv preprint arXiv:2109.06780, 2021.
> [3] Pasukonis J, Lillicrap T, Hafner D. Evaluating long-term memory in 3d mazes[J]. arXiv preprint arXiv:2210.13383, 2022.

---

> > ### Comment · Reviewer_Cf3s · 2025-11-26
> >
> > > Positional Encoding in Memory Embeddings
> >
> > I still don’t see how that can help in your case. Your memory mechanism doesn’t use temporal information: in particular, in Equation (4) you retrieve observations independent of their temporal information. So, they’re not forming a sequence in the sense of a trajectory.
> >
> > > Explanation of the “Trade-off” in Line 193
> >
> > Both of these are the same to me and are a proxy for the state.
> >
> > > Gradient Handling in Equation (4)
> >
> > Are gradients flowed back to your reads then?
> >
> > > Memory Population in Training and Inference Phases
> >
> > When memory is filled up, how do you choose to write a new cell?
> >
> > > Reason for the Absence of DreamerV3 in Figure 3
> >
> > PO-Dreamer is a modification of DreamerV3 and you already applied your method on those environments. I don’t see why you can’t remove the memory module from PO-Dreamer and run on those environments.
> >
> > > New environments
> >
> > MemoryMaze results look promising; please provide training curves as well as probing results suggested in their paper. Also, without knowing the number of seeds and the measure of reported error, I can’t tell if the results for Crafter are significant.
> >
> > Most of my main concerns are not addressed and some of them the authors didn’t reply to. Therefore, I keep my score for now.

---

> ### Author Response · Authors · 2025-11-27
>
> **W1:**
>
> Atari 100K only reflect mild partial observability. To validate PO-Dreamer’s effectiveness in harder POMDP scenarios (stronger observation ambiguity, longer-term dependencies), we conducted experiments on MemoryMaze and Crafter, we conducted experiments using a consistent set of 5 random seeds.
>
> **W2:**
>
> We conducted experiments on MemoryMaze and Crafter with a consistent set of 5 random seeds, this should resolve your confusion.
>
> **W3:**
>
> To verify the reliability of the results, we have supplemented the **confidence intervals** (calculated from experiments with 5 independent random seeds) for Atari 100K experiments as follows:
>
> ##### 1.1 Main Results on Atari 100K
> | Method       | Average ± Std| IQM|
> |--------------|------------------------|----------|
> | Ours (PO-Dreamer)   | 1.516±0.072            |0.795|
> | DIAMond      | 1.459±0.093            |0.641|
> | DreamerV3    | 1.097±0.147            |0.497|
>
> ##### 1.2 Ablation Study Results
> | Component               | Average ± Std|
> |-------------------------|------------------------|
> | Ours (PO-Dreamer) | 1.865±0.067            |
> | CE Only                 | 1.781±0.094            |
> | MG Only                 | 1.155±0.054            |
> | CO Only                 | 1.375±0.073            |
> | CO+CO                   | 1.549±0.081            |
> | W/o VAE                 | 1.621±0.072            |
> | W/o CA                  | 1.752±0.076            |
> | W/o Pos                 | 1.617±0.074            |
> | W/o Act                 | 1.817±0.080            |
>
> **W4:**
>
> The PO-Dreamer model achieves a significantly higher average score of 1.865±0.067 on Atari 100K, compared to 1.549±0.081 for CO+CO. These results confirm PO-Dreamer’s performance stems from our memory mechanism (adaptive key-frame selection + cross-attention fusion), not just the model capacity alone.
>
> **W5:**
>
> We provide a detailed quantitative comparison of parameters, FLOPs, and key computational metrics below, demonstrating that PO-Dreamer’s overhead is well-controlled and justified by its performance gains:
>
> #### Table 1: Detailed Computational Cost Comparison
> | Metric                  | DreamerV3                | DIAMOND                  | STORM                    | PO-Dreamer               |
> |-------------------------|--------------------------|--------------------------|--------------------------|--------------------------|
> | **Total Parameters**    | 14M                      | 13M                      | 25M                      | 27M                      |
> | **Parameter Breakdown** | -model_opt: 11.29M -actor_opt: 1.05M    -value_opt: 1.18M | -denoiser: 4.41M -rew_end_model: 5.90M  -actor_critic_opt: 3.23M | -model_opt: 21.71M -actor_critic_opt: 3.42M         | -model_opt: 24.96M -actor_opt: 1.05M -value_opt: 1.18M|
> | **FLOPs**      | 6.02GFLOPs                    | 29.71GFLOPs                    | 4.71GFLOPs                     | 13.48GFLOPs                    |
> |**Inference Parameter**| 9M                | 13M                  |25M                         |17M                  |
> | **Inference Latency (s)*** | 8.7±0.3 | 31.7±1.4 | 21.7±0.8 | 12.3±0.4 |
> | **Mean HNS (Performance)** | 1.097 | 1.459 | 1.266 | 1.516 |
> * Inference latency is measured as the average value obtained by running 100 steps on the NVIDIA A100.
>
> **W6:**
>
> Atari/SMAC environments have stationary transition/reward functions, but their partial observability makes the agent’s perceived state appear non-stationary (e.g., unobservable ball velocity in Pong, unseen enemy movements in SMAC). PO-Dreamer’s memory mechanism resolves this by retaining historical key frames to stable the perceived non-stationarity.
>
> To strengthen validity the performance of PO-Dreamer in more challenging partial observable scenarios, we conducted additional experiments on Crafter and MemoryMaze — two benchmarks specifically designed for testing memory and partial observability capabilities.

---

> ### Author Response · Authors · 2025-11-27
>
> Thank you for your valuable comments, which have helped improve the rigor and completeness of our work.
>
> > Positional Encoding in Memory Embeddings
>
> Historical trajectories are stored in the replay buffer and our memory mechanism maintains a sliding window to retain only the most recent memory. Positional encoding is introduced to preserve the relative temporal order of observations during memory retrieval (Equation 4), as even non-sequential retrieval requires distinguishing the temporal context of stored observations to avoid confounding recent and outdated information.
>
> > Explanation of the “Trade-off” in Line 193
>
> We appreciate your attention to this terminology. While both branches contribute to state construction and serve as proxies for the latent state, they exhibit complementary characteristics rooted in distinct feature extraction sources: **the current branch leverages local observation dynamics, and the memory-guided branch integrates global memory context.** The current observation contains substantial information for decision-making, while historical observations provide valuable cues for estimating the current state. Our ablation studies confirm that using either branch in isolation yields inferior performance compared to the combination of both branches. **If "trade-off" is not suitable, we can revise it to "collaboration".**
>
> > Gradient Handling in Equation (4)
>
> To clarify, not all gradients are propagated back to the memory reads in Equation (4). For the content in the memory window, we select key frames that are most effective for recovering the dynamic transition model. During computation, **gradients are backpropagated exclusively for these selected key frames, while no gradient propagation is performed for the unselected ones.**
>
> > Memory Population in Training and Inference Phases
>
> When the memory buffer reaches its capacity, the First-In-First-Out (FIFO) strategy is adopted: the oldest trajectory entries are discarded to accommodate new observations. This ensures that the memory maintains relevant and recent temporal information consistently during both training and inference phases.
>
> > Reason for the Absence of DreamerV3 in Figure 3
>
> We agree with the reviewer’s valuable suggestion. While we conducted ablation studies (removing the memory module) on Atari environments, DreamerV3 was initially omitted from multi-agent comparisons due to the need to adapt its single-agent RL policy to multi-agent settings. Such adaptations would require non-trivial modifications to the original DreamerV3 algorithm, and the revised version could not strictly be referred to as the standard DreamerV3. To address this gap, we supplement the ablation experiments with the **adapted DreamerV3** as a baseline for SMAC in the revised manuscript, which can refer to **B.2** in the B ADDITIONAL PARTS
>
> > New environments
>
> For MemoryMaze, we will include the requested training curves and probing results in **B.3** of the revised version, and the error bars represent the standard deviation. We further clarify that all results in these environments are reported with a fixed sequence of 5 consecutive random number seeds.

---

### Official Review · Reviewer_q9n2 · 2025-10-31

**Soundness:** 3
**Presentation:** 2
**Contribution:** 3
**Rating:** 2
**Confidence:** 4

**Summary:**

This paper introduces PO-Dreamer, which proposes an architecture to extract "important" observations to attempt to address the partial observability problem within the model-based RL framework.
Specifically, the authors propose to maintain an additional memory latent and recurrent state $(z^{mg}, h^{mg})$ to address the issue of partial observability. They propose a mechanism of *adaptive dropout* where important observations are retained by computing cosine similarity between memory embeddings and the current obs. embedding. They evaluate their performance on the ATARI 100k benchmark and on the StarCraft multi-agent challenge (SMAC) and show better performance of PO-Dreamer.

**Strengths:**

1. Most MBRL frameworks don't address partial-observability in an intelligent way (most of them employ recurrent networks) -- it is interesting to see a direction in MBRL where "relevant" observations are kept for dynamics prediction.

2. It is nice to see the open-sourcing of the code base. I've gone through it briefly (haven't run it) and the structure generally makes sense.

**Weaknesses:**

1. **[Experimental Setting]** I find the ATARI 100k experimental setting to be a bit underwhelming since ATARI is not necessarily the best benchmark for testing partial observability. This is because stacking frames in ATARI / using recurrent network/transformers (naively) can still be equally performant or, in some games, better, as clearly shown in Table 1. Instead, I'd like PO-Dreamer to be tested on more complex POMDP tasks. One good start could be the Flickering Atari Games from [1] -- perhaps rather than blanking out frames with p=0.5, you could blank a series of frames together with some probability so as to test the adaptive dropout mechanism.

Other benchmarks, such as Crafter [2], no-inventory Crafter [3], and BabyAI environments [4], are a much better suite of tasks to test PO-Dreamer on.

2. **[Analysis on the selected frames]** One of the key components of PO-Dreamer is the selection of the "key observations" using the Adaptive Dropout. A visual analysis of what frames are being selected is important. How far back is the memory encoding selecting (among the 800 frames)? This perhaps isn't super crucial for the ATARI benchmark as opposed to StarCraft or when you blank out ATARI frames [1]. It'd be interesting to see if it selects the non-blank frames or not.


-----
**References**

[1] Deep Recurrent Q-Learning for Partially Observable MDPs, Matthew Hausknecht and Peter Stone, AAAI Symposia, 2015

[2] Benchmarking the Spectrum of Agent Capabilities, Danijar Hafner, ICLR 2021

[3] Benchmarking Partial Observability in Reinforcement Learning with a Suite of Memory-Improvable Domains, Ruo Yu Tao et al., RL-C 2025

[4] https://minigrid.farama.org/environments/babyai/index.html


----
**Rationale for voting**

I'm currently voting for a rejection of the paper as my major concern is lack of experiments that have higher level of partial-observability. The core claim of the paper the memory-guided world model which can effectively address challenges of non-stationarity in POMDPs. I will base my final decision on authors' rebuttal and comments from other reviewers.

**Questions:**

3. On Table 1, STORM is marked with an asterisk that says "These methods are implemented with advanced network backbones such as Transformer and Diffusion for world models." I would like to point out that the number of transformer layers used in PO-Dreamer in the cross-attention is 3, while STORM only uses 2 layers (both have the same feature dim of 512) -- so I don't believe that STORM is a more complex model than PO-Dreamer. Please confirm if this argument is correct.


4. It is unclear to me as to what is preventing the collapse of $\hat{f_t}$ to a trivial solution? For instance, $\hat{z_t}^{mg}$ can be zeros (and so can $f_t$), and effectively this reduces to DreamerV3. It would be helpful if the authors could clarify this.

---

> ### Author Response · Authors · 2025-11-23
> **Response to AnonReviewer2(Reviewer q9n2)**
>
> We sincerely appreciate your review and valuable questions, which have helped us clarify critical details of our method. Below is a detailed response to your concerns:
>
> 1. Clarification on Model Complexity Comparison Between PO-Dreamer and STORM
>
> Thank you for pointing out the ambiguity in the complexity annotation of STORM in Table 1. We aim to clarify the key differences in the application of Transformer backbones between PO-Dreamer and STORM (as well as other starred methods) as follows:
> STORM and other baseline methods: Transformers and diffusion models are directly integrated into the core architecture of the world model, serving as the primary backbone for dynamics prediction and state modeling. This means these advanced networks are involved in both training and inference phases of the world model, leading to high computational overhead in all stages.
> PO-Dreamer: The 3-layer Transformer is only used for feature extraction in the memory fusion module (cross-attention between current observations and historical memory). It is not part of the world model’s core dynamics prediction network; instead, the world model retains the lightweight latent state modeling framework of DreamerV3.
>
> 2. Mechanisms Preventing the Model from Collapsing to a Trivial Solution (DreamerV3)
>
> We appreciate your concern about the risk of the model degenerating into DreamerV3 (e.g., memory-related latent variables becoming zero vectors). To address this issue, we designed a dual constraint mechanism combining supervised training and joint latent state modeling.
> The memory encoder’s output (latent features of historical memory) is used to supervise the decoding process of the world model. Specifically, we force the world model to reconstruct the original memory observations from the fused latent state (current observation + historical memory). If the memory-related latent variables collapse to zero, the reconstruction loss will increase sharply, penalizing the model and preventing trivial solutions. Both branches are required to collaboratively decode critical signals such as reward values and termination states during training. If either branch’s latent variables become zero, the model will fail to accurately predict these signals (as verified in our ablation study of "MG Only" (Memory Guide Only), which achieves 1.155±0.054 in Atari 100K). This joint decoding constraint ensures that both branches contribute to the latent state space, avoiding the degeneration of the memory module.
>
> We will also carefully revised the original manuscript to address all your comments, and we hope the above explanations resolve your concerns. Please feel free to let us know if you have any further questions for revision.

---

> ### Author Response · Authors · 2025-11-26
>
> Thanks for your valuable comments on the partial observability validation of our work. We would like to clarify the partial observability of the experimental environments and supplement the generalization experiments as follows:
>
> 1. Explanation of Partial Observability in Experimental Environments
> Regarding the partial observability of Atari environments, we align with the viewpoint of DRQN [1]: *"Real-world tasks often feature incomplete and noisy state information resulting from partial observability like that found in POMDPs. As Figure 1 shows, given only a single game screen, many Atari 2600 games are POMDPs. One example is the game of Pong in which the current screen only reveals the location of the paddles and the ball, but not the velocity of the ball. Knowing the direction of travel of the ball is a crucial component for determining the best paddle location."* Even with frame stacking, the dynamic state information (e.g., object velocity, hidden state transitions) makes Atari games non-fully observable scenarios.
>
> For multi-agent environments (SMAC), partial observability is manifested in two core aspects: (1) Each agent has a limited observation range and can only perceive local information rather than the global state; (2) The mutual interaction between agents leads to non-stationary environmental dynamics, where the action of one agent changes the state space perceived by others.
>
> 2. Supplementary Generalization Experiments on Specialized POMDP Benchmarks
> To verify the generalization of PO-Dreamer in more challenging partial observable scenarios, we conducted additional experiments on **Crafter** [2] and **MemoryMaze** [3]—two benchmarks specifically designed for testing memory and partial observability capabilities. The experimental results are presented below:
>
> 2.1 Crafter Experiment Results
> Crafter requires agents to collect resources, build tools, and complete complex tasks in an open world, where long-term memory of resource locations and task progress is essential.
>
> | Method       | Score       | Reward      |
> |--------------|-------------|-------------|
> | Human        | 50.5±6.8    | 14.3±2.3    |
> | PPO          | 4.6±0.3     | 4.2±1.2     |
> | Rainbow      | 4.3±0.2     | 6.0±1.3     |
> | DreamerV3    | 14.5±1.6    | 11.7±1.9    |
> | Ours (PO-Dreamer) | 16.2±1.3 | 12.3±1.4 |
>
> 2.2 MemoryMaze Experiment Results
> MemoryMaze is a 3D maze environment where agents need to remember the location of target points and navigation paths to complete tasks. We tested mazes of different sizes to evaluate long-term memory capabilities.
>
> | Method       | Memory9x9   | Memory11x11 | Memory13x13 | Memory15x15 |
> |--------------|-------------|-------------|-------------|-------------|
> | Human        | 26.4±1.6    | 44.3±2.5    | 55.5±3.5    | 67.7±4.0    |
> | IMPALA       | 23.4±0.4    | 28.2±0.3    | 27.5±0.6    | 17.7±1.4    |
> | DreamerV3    | 35.9±0.4    | 43.5±1.1    | 36.2±1.2    | 15.7±1.3    |
> | Ours (PO-Dreamer) | 38.0±1.6 | 47.6±1.5 | 50.3±2.2 | 47.7±2.7 |
>
> As shown in the results, PO-Dreamer outperforms baseline methods (including DreamerV3) on both benchmarks, especially in large-scale MemoryMaze (15x15) where long-term memory is critical—PO-Dreamer maintains stable performance, while DreamerV3 and IMPALA experience significant performance degradation. This confirms the effectiveness of our memory mechanism in handling complex partial observable scenarios.
>
>
>
>
> **References**
> [1] Hausknecht M, Stone P. Deep Recurrent Q-Learning for Partially Observable MDPs[C]//AAAI Symposia, 2015.
> [2] Hafner D. Benchmarking the spectrum of agent capabilities[J]. arXiv preprint arXiv:2109.06780, 2021.
> [3] Pasukonis J, Lillicrap T, Hafner D. Evaluating long-term memory in 3d mazes[J]. arXiv preprint arXiv:2210.13383, 2022.

---

> > ### Comment · Reviewer_q9n2 · 2025-11-26
> > **Thanks for the rebuttal. Few follow ups before my final vote.**
> >
> > Thanks to the authors for their rebuttal.
> >
> >
> > **MemoryMaze Results**: This is a really strong result and I believe this should be the main focus of the experiments of this paper. I'm not convinced on the level of partial observability of Atari 100k and the argument that the authors provided in the rebuttal is not enough. For instance they quote DRQN:
> >
> > > Even with frame stacking, the dynamic state information (e.g., object velocity, hidden state transitions) makes Atari games non-fully observable scenarios.
> >
> > This can be true for a few Atari games but not for the Pong. Specifically, for Pong, frame stacking of past 2 frames would give me velocity and acceleration of each of the objects, making it fully observable!
> >
> > I'd recommend all the ablations be performed on the MemoryMaze.
> >
> >
> > **Clarification on Model Complexity Comparison Between PO-Dreamer and STORM**
> >
> > > This means these advanced networks are involved in both training and inference phases of the world model, leading to high computational overhead in all stages.
> >
> > Isn't it the case for PO-Dreamer as well that the cross attention needs to be performed in both training and inference phases to obtain $s_t$?
> >
> > **Mechanisms Preventing the Model from Collapsing to a Trivial Solution (DreamerV3)**
> >
> > I'm still confused about this. So currently in PO-Dreamer, $\hat{f_t}$ is trained with a latent "reconstruction" loss on $f_t$ which is obtained as the output of the memory encoder. My question was what in this pipeline is preventing from $f_t$ going to a trivial solution -- meaning that only the $CO$ part of the PO-Dreamer is active. $s_t$ still learns some meaningful representation since it has to decode the input observation $\hat{o_t}$ which is supervised by observation reconstruction loss.
> >
> > **Another quick question**: What losses aren't present in the "MG Only" experiment?

---

> ### Author Response · Authors · 2025-11-28
>
> Thank you for your valuable feedback.
>
> > MemoryMaze Results and suggestion
>
> We fully agree that the observability of Atari environments is a critical point worthy of in-depth consideration. As you insightfully noted, the partial observability characteristics may not be sufficiently prominent in standard Atari 100k settings.
> To address this concern and align with your recommendation, we will supplement comprehensive ablation studies on the  **MemoryMaze** environment (**Chapter B.1** in Appendix B of the newly submitted revised version). These experiments will systematically verify the effectiveness of our proposed mechanism under representative partial observability conditions.
>
> >Clarification on Model Complexity Comparison Between PO-Dreamer and STORM
>
> You are correct that the cross-attention mechanism between our two branches is indeed involved in both training and inference phases of PO-Dreamer—this design is essential for fusing current observation and memory. While this introduces additional computational, we have quantified and validated that the overhead is well-controlled and justified by significant performance gains, as shown in Table 1:
>
> Table 1 below provides a comprehensive breakdown of computational metrics for full transparency:
>
> | Metric                  | DreamerV3       | DIAMOND         | STORM           | PO-Dreamer      |
> |-------------------------|-----------------|-----------------|-----------------|-----------------|
> | Total Parameters        | 14M             | 13M             | 25M             | 27M             |
> | Parameter Breakdown     | -model_opt: 11.29M -actor_opt: 1.05M -value_opt: 1.18M | -denoiser: 4.41M -rew_end_model: 5.90M -actor_critic_opt: 3.23M | -model_opt: 21.71M -actor_critic_opt: 3.42M | -model_opt: 24.96M -actor_opt: 1.05M -value_opt: 1.18M |
> | FLOPs                   | 6.02 GFLOPs     | 29.71 GFLOPs    | 4.71 GFLOPs     | 13.48 GFLOPs     |
> | Inference Parameter     | 9M              | 13M             | 25M             | 17M             |
> | Inference Latency (s)*  | 8.7±0.3         | 31.7±1.4        | 21.7±0.8        | 12.3±0.4        |
> | Mean HNS (Performance)  | 1.097           | 1.459           | 1.266           | 1.516           |
>
> *Note: Inference latency is measured as the average value obtained by running 100 steps on the NVIDIA A100.*
>
> In summary, while PO-Dreamer’s cross-attention mechanism contributes to computational overhead in both training and inference, our quantitative analysis confirms that this overhead is tightly controlled through architectural optimizations. The superior performance across benchmarks validates that the trade-off between complexity and effectiveness is well-balanced, making PO-Dreamer both expressive and practically feasible.
>
> > Mechanisms Preventing the Model from Collapsing to a Trivial Solution (DreamerV3)
>
> Thank you for your insightful question regarding the mechanisms preventing the model from collapsing to a trivial solution. We designed a cross-attention mechanism for the two branches (the current observation branch and memory-guided branch) to enable information sharing and feature coupling between them: First, the two branches extract features from current observations and historical observations, respectively, and jointly form the complete latent state of the world model $s_t = CAT(h_t^{co}, z_t^{co}, h_t^{mg}, z_t^{mg}$, which is used to predict observations, reward and continue flag—omitting either branch will lead to failures in downstream decoding tasks (e.g., inaccurate observation prediction or reward prediction), and this constraint supervises both branches to learn effective features. Second, the tight coupling of the latent states $h_t^{co}$ and $h_t^{mg}$ from the two branches via cross-attention ensures that the degradation of either branch will impair the accuracy of the dynamic transition model constructed by the other branch, thereby forcing the degraded branch to learn meaningful representations for building an accurate state transition model.
>
> In summary, the branch coupling induced by the cross-attention mechanism (jointly forming the world model) and the supervision constraint from downstream prediction tasks form a dual guarantee, which effectively avoids the emergence of trivial solutions and ensures that both branches learn complementary and meaningful features.
>
>
>
> > What losses aren't present in the "MG Only" experiment?
>
> As for "MG Only", the prediction losses calculation follows the same approach as PO-Dreamer in terms of the four predictive losses: observation, memory, reward and continue losses. However, **the dynamics loss and representation loss associated with the current observation branch is not included**($\text{KL}[\text{sg}(z_t^{co}) \parallel \hat{z}_t^{co}]$ and $\text{KL}[z_t^{co} \parallel \text{sg}(\hat{z}_t^{co}）]$ aren't present).

---

### Official Review · Reviewer_GV13 · 2025-11-01

**Soundness:** 3
**Presentation:** 3
**Contribution:** 3
**Rating:** 6
**Confidence:** 4

**Summary:**

This submission proposes PO-Dreamer, a memory-guided world model for partially observable reinforcement learning. The key proposa is a dual-encoder architecture that separately processes current observations and historical memory, then fuses them through attention mechanisms to predict state transitions and rewards. The memory encoder uses adaptive dropout to select relevant frames from a memory window based on cosine similarity, and models memory features as a Gaussian distribution via VAE. Experiments on Atari 100K and SMAC demonstrate performance improvement and strong results on multi-agent tasks.

**Strengths:**

1. The submission is generally well-written with clear figures and good organization.
2. The experiment is comprehensive, showing improved performance across diverse Atari and multi-agent benchmark tasks and compared against a wide array of strong MBRL baselines.
3. The proposed architecture is properly motivated to addresses partial observability by explicitly modeling memory alongside current observations and fusion mechanism using attention.
4. Comprehensive ablation studies were perform to validate each component choice and hyperparameter sensitivity.

**Weaknesses:**

1. Performance results is not reported with confidence interval, making it hard to assess whether the difference is meaningful or within the range of variance. Please include confidence interval for both main results and ablation studies.
2. Computational cost was not analyzed: No discussion of wall-clock time, memory usage, or training efficiency compared to baselines. How is the computational overhead compared to methods like Dreamer-v3. Given τ=800 and attention mechanisms, computational overhead could be significant.
3. There is little analysis on when memory helps. The submission claims benefits for adding memory but provides limited systematic analysis of which task characteristics benefit from memory. Can you discuss when memory provides the largest benefits? What task characteristics correlate with memory usefulness? For instance, in atari games, it seems like baseline models perform even better in many games.
4. Limitation of the work is not discussed.

**Questions:**

1. Cross-episode vs. current episode: Given "CE Only" performs reasonably well (1.781, vs. 1.865 for the full model), how much value does cross-episode memory actually add? Is the 800-step window typically within a single episode for most games?
2. Is there any way to measure whether the world model learned by the proposed PO-Dreamer is better than other method say Dreamer-v3? Is this the major driver for improved policy learning? Or something else?
3. The cosine similarity-based memory selection seems relatively simple and arbitrary, and not well justified against alternatives.
4. Line 72 typo: "environmentment's" → "environment's"

---

> ### Author Response · Authors · 2025-11-23
> **Response to AnonReviewer1(Reviewer GV13)**
>
> We sincerely appreciate your insightful comments and constructive suggestions, which have significantly helped us improve the quality of our paper. Below is a detailed response to your concerns, along with supplementary experimental results and revisions:
>
> ####1. Supplementary Confidence Intervals for Experimental Results
>
> ##### 1.1 Main Results on Atari 100K
> | Method       | Average ± Std| IQM|
> |--------------|------------------------|----------|
> | Ours (PO-Dreamer)   | 1.516±0.072            |0.795|
> | DIAMond      | 1.459±0.093            |0.641|
> | DreamerV3    | 1.097±0.147            |0.497|
>
> ##### 1.2 Ablation Study Results
> | Component               | Average ± Std|
> |-------------------------|------------------------|
> | Ours (PO-Dreamer) | 1.865±0.067            |
> | CE Only                 | 1.781±0.094            |
> | MG Only                 | 1.155±0.054            |
> | CO Only                 | 1.375±0.073            |
> | CO+CO                   | 1.549±0.081            |
> | W/o VAE                 | 1.621±0.072            |
> | W/o CA                  | 1.752±0.076            |
> | W/o Pos                 | 1.617±0.074            |
> | W/o Act                 | 1.817±0.080            |
>
>
> #### 2. Value of Cross-Episode Memory vs. Intra-Episode Memory
> ##### 2.1 Quantitative Analysis
> The full model (1.865±0.067) achieves a 4.7% performance improvement over the CE Only variant (1.781±0.094). Although the absolute improvement seems modest, PO-Dreamer achieves state-of-the-art (SOTA) performance in 6 out of 8 ablation study scenarios with cross-episode memory incorporated. This demonstrates that cross-episode memory is a key contributor to the method’s success in complex tasks.
>
> ##### 2.2 Qualitative Differences
> Intra-episode memory only addresses *information gaps within a single episode* (e.g., retaining short-term action history). In contrast, cross-episode memory enables the model to learn *cross-episode common patterns* (e.g., the frequent formation strategies of enemies in SMAC). This design reduces exploratory costs and enhances decision-making robustness in non-stationary environments.
>
> ##### 2.3 Rationality of the 800-Step Memory Window
> **Atari environments**: Most games have an average intra-episode step count of ~500 steps.
> **SMAC environments**: The average intra-episode step count is ~300 steps.
> (PS: Both of them with even shorter steps in early training stages)
>
> An 800-step window covers most of the critical historical information in a single episode for both environments. We validated the performance of different window lengths via ablation experiments (Figure 4(a) in the paper), confirming that 800 steps achieves an optimal balance between computational cost and performance.
>
> #### 3. Validation of the World Model’s Superiority and Performance Attribution
> The performance improvement of PO-Dreamer over DreamerV3 primarily stems from **enhanced accuracy of the world model**. As shown in Appendix A4 (Loss Comparison), our method achieves significantly lower state reconstruction loss and reward prediction loss than DreamerV3, indicating a more precise modeling of environmental dynamics.
>
> Notably, the reinforcement learning policy of PO-Dreamer is identical to that of DreamerV3 in Atari environments, so policy differences do not contribute to the performance gap. We further analyzed other potential factors (e.g., model capacity, memory fusion mechanisms) in ablation studies, confirming that the improved world model is the dominant driver of performance.
>
> #### 4. Rationale for Cosine Similarity-Based Memory Selection
> The core goal of the memory selection mechanism is to **extract key frames from long historical memory for feature extraction**, thereby improving the model’s decision-making based on current observations. Cosine similarity is chosen as the selection criterion for two key reasons:  (1)It effectively measures the *semantic relevance* between the embedding of the current observation and historical frames (e.g., frames containing "key items" in Atari games have high similarity to frames containing "target doors").  (2)It filters out redundant frames that provide no additional information, reducing computational overhead while preserving critical memory content.
>
> #### 5. Correction of Spelling Errors
> We will correct the typo ("environmentment's" → "environment's") at Line 72 in the revised manuscript.
>
> We hope the above responses address your concerns. Please let us know if you have any further questions or suggestions for revision.

---

> ### Author Response · Authors · 2025-11-26
>
> Thanks for your valuable comments on the partial observability validation of our work. We would like to clarify the partial observability of the experimental environments and supplement the generalization experiments as follows:
>
> 1. Explanation of Partial Observability in Experimental Environments
> Regarding the partial observability of Atari environments, we align with the viewpoint of DRQN [1]: *"Real-world tasks often feature incomplete and noisy state information resulting from partial observability like that found in POMDPs. As Figure 1 shows, given only a single game screen, many Atari 2600 games are POMDPs. One example is the game of Pong in which the current screen only reveals the location of the paddles and the ball, but not the velocity of the ball. Knowing the direction of travel of the ball is a crucial component for determining the best paddle location."* Even with frame stacking, the dynamic state information (e.g., object velocity, hidden state transitions) makes Atari games non-fully observable scenarios.
>
> For multi-agent environments (SMAC), partial observability is manifested in two core aspects: (1) Each agent has a limited observation range and can only perceive local information rather than the global state; (2) The mutual interaction between agents leads to non-stationary environmental dynamics, where the action of one agent changes the state space perceived by others.
>
> 2. Supplementary Generalization Experiments on Specialized POMDP Benchmarks
> To verify the generalization of PO-Dreamer in more challenging partial observable scenarios, we conducted additional experiments on **Crafter** [2] and **MemoryMaze** [3]—two benchmarks specifically designed for testing memory and partial observability capabilities. The experimental results are presented below:
>
> 2.1 Crafter Experiment Results
> Crafter requires agents to collect resources, build tools, and complete complex tasks in an open world, where long-term memory of resource locations and task progress is essential.
>
> | Method       | Score       | Reward      |
> |--------------|-------------|-------------|
> | Human        | 50.5±6.8    | 14.3±2.3    |
> | PPO          | 4.6±0.3     | 4.2±1.2     |
> | Rainbow      | 4.3±0.2     | 6.0±1.3     |
> | DreamerV3    | 14.5±1.6    | 11.7±1.9    |
> | Ours (PO-Dreamer) | 16.2±1.3 | 12.3±1.4 |
>
> 2.2 MemoryMaze Experiment Results
> MemoryMaze is a 3D maze environment where agents need to remember the location of target points and navigation paths to complete tasks. We tested mazes of different sizes to evaluate long-term memory capabilities.
>
> | Method       | Memory9x9   | Memory11x11 | Memory13x13 | Memory15x15 |
> |--------------|-------------|-------------|-------------|-------------|
> | Human        | 26.4±1.6    | 44.3±2.5    | 55.5±3.5    | 67.7±4.0    |
> | IMPALA       | 23.4±0.4    | 28.2±0.3    | 27.5±0.6    | 17.7±1.4    |
> | DreamerV3    | 35.9±0.4    | 43.5±1.1    | 36.2±1.2    | 15.7±1.3    |
> | Ours (PO-Dreamer) | 38.0±1.6 | 47.6±1.5 | 50.3±2.2 | 47.7±2.7 |
>
> As shown in the results, PO-Dreamer outperforms baseline methods (including DreamerV3) on both benchmarks, especially in large-scale MemoryMaze (15x15) where long-term memory is critical—PO-Dreamer maintains stable performance, while DreamerV3 and IMPALA experience significant performance degradation. This confirms the effectiveness of our memory mechanism in handling complex partial observable scenarios.
>
>
>
>
> **References**
> [1] Hausknecht M, Stone P. Deep Recurrent Q-Learning for Partially Observable MDPs[C]//AAAI Symposia, 2015.
> [2] Hafner D. Benchmarking the spectrum of agent capabilities[J]. arXiv preprint arXiv:2109.06780, 2021.
> [3] Pasukonis J, Lillicrap T, Hafner D. Evaluating long-term memory in 3d mazes[J]. arXiv preprint arXiv:2210.13383, 2022.

---

### Author Response · Authors · 2025-11-23
**Revised Response to Reviewers**

Thanks for your valuable comments on the partial observability validation of our work. We would like to clarify the partial observability of the experimental environments and supplement the generalization experiments as follows:

1. Explanation of Partial Observability in Experimental Environments
Regarding the partial observability of Atari environments, we align with the viewpoint of DRQN [1]: *"Real-world tasks often feature incomplete and noisy state information resulting from partial observability like that found in POMDPs. As Figure 1 shows, given only a single game screen, many Atari 2600 games are POMDPs. One example is the game of Pong in which the current screen only reveals the location of the paddles and the ball, but not the velocity of the ball. Knowing the direction of travel of the ball is a crucial component for determining the best paddle location."* Even with frame stacking, the dynamic state information (e.g., object velocity, hidden state transitions) makes Atari games non-fully observable scenarios.

For multi-agent environments (SMAC), partial observability is manifested in two core aspects: (1) Each agent has a limited observation range and can only perceive local information rather than the global state; (2) The mutual interaction between agents leads to non-stationary environmental dynamics, where the action of one agent changes the state space perceived by others.

2. Supplementary Generalization Experiments on Specialized POMDP Benchmarks
To verify the generalization of PO-Dreamer in more challenging partial observable scenarios, we conducted additional experiments on **Crafter** [2] and **MemoryMaze** [3]—two benchmarks specifically designed for testing memory and partial observability capabilities. The experimental results are presented below:

2.1 Crafter Experiment Results
Crafter requires agents to collect resources, build tools, and complete complex tasks in an open world, where long-term memory of resource locations and task progress is essential.

| Method       | Score       | Reward      |
|--------------|-------------|-------------|
| Human        | 50.5±6.8    | 14.3±2.3    |
| PPO          | 4.6±0.3     | 4.2±1.2     |
| Rainbow      | 4.3±0.2     | 6.0±1.3     |
| DreamerV3    | 14.5±1.6    | 11.7±1.9    |
| Ours (PO-Dreamer) | 16.2±1.3 | 12.3±1.4 |

2.2 MemoryMaze Experiment Results
MemoryMaze is a 3D maze environment where agents need to remember the location of target points and navigation paths to complete tasks. We tested mazes of different sizes to evaluate long-term memory capabilities.

| Method       | Memory9x9   | Memory11x11 | Memory13x13 | Memory15x15 |
|--------------|-------------|-------------|-------------|-------------|
| Human        | 26.4±1.6    | 44.3±2.5    | 55.5±3.5    | 67.7±4.0    |
| IMPALA       | 23.4±0.4    | 28.2±0.3    | 27.5±0.6    | 17.7±1.4    |
| DreamerV3    | 35.9±0.4    | 43.5±1.1    | 36.2±1.2    | 15.7±1.3    |
| Ours (PO-Dreamer) | 38.0±1.6 | 47.6±1.5 | 50.3±2.2 | 47.7±2.7 |

As shown in the results, PO-Dreamer outperforms baseline methods (including DreamerV3) on both benchmarks, especially in large-scale MemoryMaze (15x15) where long-term memory is critical—PO-Dreamer maintains stable performance, while DreamerV3 and IMPALA experience significant performance degradation. This confirms the effectiveness of our memory mechanism in handling complex partial observable scenarios.




**References**
[1] Hausknecht M, Stone P. Deep Recurrent Q-Learning for Partially Observable MDPs[C]//AAAI Symposia, 2015.
[2] Hafner D. Benchmarking the spectrum of agent capabilities[J]. arXiv preprint arXiv:2109.06780, 2021.
[3] Pasukonis J, Lillicrap T, Hafner D. Evaluating long-term memory in 3d mazes[J]. arXiv preprint arXiv:2210.13383, 2022.

---

### Note · Authors · 2026-01-05

I have read and agree with the venue's withdrawal policy on behalf of myself and my co-authors.